# UNIVERSAL ALGORITHM-IMPLICIT LEARNING

## ABSTRACT

Current meta-learning methods are constrained to narrow task distributions with fixed feature and label spaces, limiting applicability. We present TAIL, a novel algorithm-implicit meta-learner that functions across tasks with varying domains, modalities, and label configurations. Our approach reformulates the few-shot learning problem as a sequence modeling problem. We train a non-causal transformer on sequences of data-label-pairs and and unlabeled query sample, to directly predict the label of the query sample. This causes the transformer to learn an implicit learning algorithm, which enables it to learn new concepts at test time without fine-tuning. Empirically, TAIL achieves state-of-the-art performance on standard benchmarks while generalizing to unseen domains and modalities. Unlike other meta-learning methods, it sustains strong performance on tasks with up to $20\times$ more classes than in training while providing orders of magnitude computational savings. Moreover, we introduce a theoretical framework for meta-learning, which allows us to formally describe important properties of meta-learning paradigms.

R4.5

## 1 INTRODUCTION

Modern deep learning has achieved remarkable success but typically depends on massive datasets and heavy computation. In many real-world settings, however, collecting large labeled datasets is costly, ethically constrained, or infeasible. Meta-learning, or "learning to learn," addresses this challenge by training algorithms to rapidly adapt to new tasks from only a few examples.

Meta-learning can evaluated under two settings. In the *in-domain* case, the meta-learner is trained and tested on tasks from the same domain, measuring how quickly it can adapt to new but related problems. In contrast, some more recent works explore the more demanding *cross-domain* setting, where training and target tasks stem from different domains. Here, knowledge acquired in the source domain may not transfer directly, requiring the learner to develop generalizable learning strategies rather than overfitting to domain-specific features. A true "learning-to-learn" algorithm should therefore succeed even when the target domain is entirely different from the source (e.g. a model learned on images should still work on text). Ultimately, the ambition of cross-domain meta-learning can be described as achieving *practical universality*, i.e. functioning as a robust learning algorithm across task distributions that vary in feature domains, label spaces, and loss functions.

Many prior meta-learning approaches fail in one or more aspects of practical universality. Recently, it has been shown that many meta-learning algorithms are limited when there is a large shift between the feature domain of the meta-training data and the application dataset (Chen et al., 2020; Luo et al., 2023; Guo et al., 2020; Oh et al., 2022; Hospedales et al., 2021; Vettoruzzo et al., 2024). We hypothesize that most methods suffer from structural limitations which prevent cross-domain and cross-modal generalization. Moreover, there is currently no theoretical framework for meta-learning that provides the right taxonomy to surface such structural limitations of current meta-learning methods.

R3.1

In this paper, we address both of these issues. First, we introduce a theoretical framework for meta-learning, which allows us to formally describe important properties of meta-learning paradigms. We propose a distinction between *algorithm-explicit* and *algorithm-implicit* learning systems, which proves to be a key difference between meta-learning algorithms that generalize well and those that do not. Moreover, our framework establishes the notion of practical universality, which provides a formalization of an algorithm's generalization ability to different tasks.

Table 1: Sequence-based meta-learners compared in dimensions relevant to practical universality.

| Method | Causal architecture? | Variable Feature Spaces | Variable Label Spaces | Flexible Sequence Length | Key Limitation |
|---|---|---|---|---|---|
| **SNAIL** (Mishra et al., 2018) | Causal | $\times$ | $\times$ | $\times$ | Cannot generalize across modalities, label spaces or support set size |
| **GPICL** (Kirsch et al., 2024) | Causal | theoretically | $\times$ | $\times$ | Cannot generalize across label spaces or support set size; no cross-modality experiments |
| **CAML** (Fifty et al., 2023) | Non-causal | $\times$ | $\times$ | $\checkmark$ | Cannot generalize across modalities or label spaces |
| **TAIL** (ours) | **Non-causal** | $\checkmark$ | $\checkmark$ | $\checkmark$ | — |

R2.1

Second, we present a novel *algorithm-implicit* meta-learning method based on transformers that makes substantial advances towards practical universality (see Fig. 1). Similar to previous works (Santoro et al., 2016; Kaiser et al., 2017; Kirsch et al., 2024; Fifty et al., 2023), we reformulate the few-shot learning problem as a sequence modeling problem. We train a non-causal transformer on sequences of data-label-pairs and and unlabeled query sample, to directly predict the label of the query sample. This causes the transformer to learn an implicit learning algorithm, which enables it to learn new concepts at test time without fine-tuning.

While the paradigm of sequence-based meta-learning has been established many years, the community has been continually improving this idea. However, prior approaches were limited to toy datasets, single domains or single modalities and did not generalize well across domains, modalities and arbitrary numbers of classes. Table 1 compares recent sequence-based methods. To our knowledge, our approach is the first model-based meta-learner to succeed simultaneously across domain, modality, and label cardinality shifts. We address three critical challenges that have prevented previous meta-learning methods from achieving this goal. **(i) Universal feature handling:** We develop a feature encoding strategy that combines task-specific encoders with randomly sampled projections into a common latent space, enabling seamless transfer across completely different modalities (images, text, medical scans) without architectural modifications and without retraining. **(ii) Universal label handling:** We introduce a randomized global dictionary of learnable embeddings, allowing the model to handle arbitrary label sets and extrapolate to tasks with more classes than seen during training. **(iii) Computational efficiency:** Our transformer-based method scales to tasks with much larger label sets while maintaining strong performance and requiring only a fraction than previous transformer-based methods.

R2.1

Our method sets new state-of-the-art results on diverse few-shot classification tasks and generalizes to unseen domains and modalities. It retains strong performance on tasks with up to $20\times$ more classes than in training, a capability unmatched by existing meta-learning methods. We demonstrate that algorithm-implicit approaches outperform algorithm-explicit ones for small support sets and varied tasks.

R2.3

## 2 PROBLEM FORMULATION

Here, we formalize the distinction between algorithm-explicit and algorithm-implicit learning and practical universality, and introduces a theoretical framework for evaluating few-shot algorithms.

### 2.1 THE LEARNING PROBLEM

We formalize a learning task as $T \coloneqq (\mathcal{X}, \mathcal{Y}, p, \ell)$, where $\mathcal{X}$ is the feature domain or input domain, $\mathcal{Y}$ is the label domain or output domain, $p(x, y)$ is a distribution of data with $x \in \mathcal{X}, y \in \mathcal{Y}$, and $\ell : \mathcal{Y} \times \mathcal{Y} \to \mathbb{R}$ is a loss function, which measures a "distance" between a predicted value and the ground truth label. $\ell$ is assumed to be measurable, non-negative, but not necessarily a true metric.

The learning problem on task $T$ consists of finding a function $f : \mathcal{X} \to \mathcal{Y}$, which is typically called "hypothesis" or "model", that minimizes the risk

$$R(f) = \underset{(x,y)\sim p(x,y)}{\mathbb{E}} \left[ \ell(f(x), y) \right].$$

In practice, the true data distribution is unknown, and $f$ is estimated using a sample from $p$ called the training set, or support set $S$ using supervised learning.

Formally, given the space $\mathcal{D} := \mathrm{supp}(p) \subseteq \mathcal{X} \times \mathcal{Y}$ of possible data, defined by the support of the distribution $p$, we can define a support set $S \subset \mathcal{D}$ as $S := \{(x_i, y_i)\}_{i=1}^{|S|}$ with $(x_i, y_i) \in \mathrm{supp}(p)$. Based on this notation we can define the following:

**Definition 1** (Learning Algorithm). A learning algorithm $\mathcal{A} : \mathcal{P}(\mathcal{X} \times \mathcal{Y}) \to \mathcal{Y}^{\mathcal{X}}, \ S \mapsto f$ is a function that maps a dataset $S \subseteq \mathcal{D}$ to a hypothesis $f$.

## 2.2 META-LEARNING

Meta-learning can be understood as finding a learning algorithm $\mathcal{A}$ through meta-optimization over the space $\mathbb{A}$ of possible learning algorithms. In meta-learning we assume access to a meta-dataset $\mathcal{T}$ of tasks sampled from a distribution of tasks $\nu$. Formally, the meta-learning problem on a meta-dataset $\mathcal{T}$ can be described as finding a learning algorithm $\mathcal{A}$ that minimizes $\quad$ **R2.3**

$$R^{\mathrm{Meta}}(\mathcal{A}) = \mathop{\mathbb{E}}_{T \sim \nu} \mathop{\mathbb{E}}_{S \sim \bigcup_{n \geq 1} p_T^n} R_T(\mathcal{A}(S)) = \mathop{\mathbb{E}}_{T \sim \nu} \mathop{\mathbb{E}}_{S \sim p_T^n} \mathop{\mathbb{E}}_{(x,y) \sim p_T} [\ell_T(\mathcal{A}(S)(x), y)] \ . \qquad \textbf{R2.3}$$

## 2.3 ALGORITHM-EXPLICIT VS ALGORITHM-IMPLICIT LEARNING

Rather than first learning a hypothesis and using it for prediction, one can directly make predictions for a data point conditioned on a support set, without explicitly computing a hypothesis $f$. We refer to such a mapping as a *demonstration-conditioned inference (DCI) function*.

**Definition 2** (Demonstration-Conditioned Inference (DCI)). A DCI function

$$g : \mathcal{P}(\mathcal{X} \times \mathcal{Y}) \times \mathcal{X} \to \mathcal{Y}, \quad (S, x) \mapsto \hat{y}$$

is a function that directly maps a support set $S \subseteq \mathcal{D}$ and query point $x$ to a prediction $\hat{y}$.

Note that for any deterministic DCI function $g$ and fixed $S$, there exists an induced hypothesis $f_S$ where $f_S(x) = g(S, x)$. That is, $g$ implicitly defines a learning algorithm $\mathcal{A}_g$ where $\mathcal{A}_g(S) = f_S$. Vice-versa, an algorithm $\mathcal{A}$ induces a DCI $g_{\mathcal{A}}$ with $g_{\mathcal{A}}(S, x) = \mathcal{A}(S)(x)$.

We introduce a fundamental distinction between learning paradigms based on whether the learning algorithm is explicitly specified or implicitly emerges.

**Algorithm-Explicit Learning:** Intuitively, a learning system is *algorithm-explicit* if its training procedure is explicitly specified. We formally define such a system, as one which is characterized by an explicit procedure $\mathcal{A}$ that maps a dataset $S$ to a hypothesis $f$ (or equivalently $g$), i.e. $\mathcal{A}(S) = f$. A good example for algorithm-explicit learning are MLPs optimized by (stochastic) gradient descent. The learning algorithm $\mathcal{A}$ on a training set $S$ is defined as iteratively updating the MLP's parameters $\theta$ with GD steps on samples from $S$, yielding the hypothesis $f_\theta$. Other examples of explicitly defined learning systems are $k$-nearest neighbors (Cover & Hart, 1967) ($\mathcal{A}$ stores $S$ and $f$ performs distance-based voting), or MAML (Finn et al., 2017) ($\mathcal{A}$ performs $k$ steps of gradient descent from initialization $\theta_0$ to find $f$).

**Algorithm-Implicit Learning:** A learning system is *algorithm-implicit* if it operates through a parameterized DCI function $g_\theta$ where the learning algorithm $\mathcal{A}_g$ emerges from the learned parameters $\theta$ but is never explicitly specified. The implicit $\mathcal{A}_g$ is defined only through the behavior of $g_\theta(S, \cdot)$ for various $S$. This means that the implicit learning algorithm $\mathcal{A}$ is a black box with little if any inductive biases. Examples of algorithm-implicit learning are attention-based meta-learners, such as SNAIL (Mishra et al., 2018), CAML (Fifty et al., 2023) and GPICL (Kirsch et al., 2024). Another example is in-context learning (Brown et al., 2020; Wu et al., 2025). $\quad$ **R4.2**

Algorithm-explicit approaches have externally specified rules and may only learn a narrow set of parameters. The resulting strong inductive biases help when meta-training data are scarce. In contrast, algorithm-implicit approaches place no assumptions on the learning algorithm, allowing the model to learn more flexible and powerful algorithms, but requiring more meta-training tasks to do so. Any inductive biases come only from the computational structure of $g$, not from explicit design constraints of the learning algorithm. An analogy is the evolution from manual feature engineering to deep learning: in the feature-engineering era, models relied on human-designed biases to perform reasonably well with limited data. Deep learning introduced far weaker inductive biases but greater representational power, at the cost of requiring more data to exploit that power. We believe learning

algorithms can undergo a similar shift, which will allow meta-learners to handle arbitrary domains and modalities.                                                    R2.5, R4.3

Our method follows an algorithm-implicit approach, meta-training parameters $\theta$ so that a $g_\theta$ function learns to process support sets without an explicit algorithm.

## 2.4 PRACTICAL UNIVERSALITY AND UNIVERSAL LEARNING ALGORITHMS

Traditional learning theory provides notions of universality that are asymptotic in nature.

**Definition 3** (Universal Consistency). A learning algorithm $\mathcal{A}$ is *consistent* with respect to a certain distribution $p$ over $\mathcal{X} \times \mathcal{Y}$ if the risk of the model $\mathcal{A}(S_n)$ converges to the Bayes risk $R^* = \inf_f R(f)$, as the size of the support set $n = |S_n| \to \infty$ where $S_n \sim p^n$, i.e. if

$$\lim_{n \to \infty} R(\mathcal{A}(S_n)) = R^* \,.$$

A learning algorithm $\mathcal{A}$ is *universally consistent* if it is consistent for any distribution $p$.

However, universal consistency makes no claims about finite-sample performance. To be able to analyze the finite sample performance of an algorithm, we formalize the idea of a *learning curve*.

**Definition 4** (Learning Curve). The learning curve $\alpha_T$ of an algorithm $\mathcal{A}$ on a task $T$ computes the expected residual risk conditioned on the size of the support set. It is defined as

$$\alpha_T(\mathcal{A}, n) = \mathbb{E}_{S \sim p_T^n}[R_T(\mathcal{A}(S)) - R_T^*] \,.$$

Intuitively, we would like a learning algorithm to have a lower residual risk with an increasing amount of training data. With this, we can now define the notion of a *valid learning algorithm*, analogous to the asymptotic notion of consistency.

**Definition 5** (Valid Learning Algorithm). An algorithm $\mathcal{A}$ qualifies as a *valid learning algorithm* for task $T$ if $\alpha_T(\mathcal{A}, n)$ is monotonically non-increasing in $n$ and for any $\varepsilon > 0$ there exists an $n$ with $\alpha_T(a, n) < \varepsilon$.

**Definition 6** (Practical Universality). A learning algorithm $\mathcal{A}$ (or a DCI $g_\theta$ inducing an algorithm $\mathcal{A}_g$) is practically universal with respect to a distribution of tasks $\nu$ if it is a valid learning algorithm on any task $T$ in the class of tasks $\mathrm{supp}(\nu)$.

In this work, we consider tasks with varying feature domains and label domains. Typically, in meta-learning research, all tasks in the meta-dataset $T \in \mathcal{T}$ are considered to have the same feature space, label space and loss function. Most papers implicitly assume that $\mathcal{Y}_T \cong \{1, \cdots, k\}$ for some fixed number of classes $k$. Instead, here we allow tasks to have feature spaces label spaces and data distributions that differ from each other. Importantly, a test time task $T' \in \mathcal{T}_{\text{test}}$ might have a feature space or label space that is not represented in $\mathcal{T}_{\text{train}}$. We classify an algorithm $\mathcal{A}$ as a universal learning algorithm only if it also qualifies as a learning algorithm on such test time tasks.

## 2.5 FEW-SHOT BENCHMARKING

Intuitively, a good few-shot algorithm needs to learn a new task using only a limited number of labeled examples. In practice, tasks are presented as a pair $(S, Q)$ of support and query sets and not with their underlying data distribution. For each *episode* of $N$-shot learning on a task $T$, we sample a support set $S \subset \mathcal{D}_T = \mathrm{supp}(p_T)$ such that $S$ contains $N$ i.i.d. samples from $p_T(x \mid y)$ for each label $y \in \mathcal{Y}$ in the task. We then sample a query set $Q \subset \mathcal{D} \setminus S$ such that it contains a fixed number $N_Q$ of i.i.d. query samples from $p_T(x \mid y)$ for each label $y \in \mathcal{Y}$ in the task. We write $S \sim p_T^n$ and $Q \sim p_T^{n'}$ with $n := N \cdot |\mathcal{Y}| = |S|$ and $n' := N_Q \cdot |\mathcal{Y}| = |Q|$. We call the resulting pair $(S, Q)$ an $N$-shot instance of task $T$. Evaluating algorithms on many $N$-shot episodes sampled from our distribution of tasks can is used to estimate $\mathbb{E}_{T \sim \nu} \, \alpha_T(\mathcal{A}, n)$ with $n = N \cdot |LD_T|$.

## 3 A TRANSFORMER-BASED UNIVERSAL ALGORITHM-IMPLICIT LEARNER

As discussed in the previous section, a demonstration-conditioned inference function can be a parametrized black-box function $g_\theta$ and the parameters $\theta$ of such a function can be meta-learned

using tasks from a meta-training set. We choose to implement $g_\theta$ using a non-causal transformer that processes support and query examples jointly to directly produce predictions for the query, which we coin the Transformer-based Algorithm-Implicit Learner (TAIL). This approach does not require test-time training and makes predictions using only a single forward pass, allowing for efficient deployment under computational constraints.

Each element of the input sequence to the transformer represents a sample from the support set, including its label, or an unlabeled query sample. For a support set $S = \{(x_i, y_i)\}_{i=1}^n$ with $n := |S| = N \cdot K$ and a query sample $x'$ from $Q$, the input sequence is given by $Z = (z_1, \ldots, z_n, z')$. In practice, we process all query samples together in one sequence for better training efficiency. This speeds up training and testing by orders of magnitude and we experimentally show the computational advantages. For simplicity we will continue to use the former notation with a single query sample. The transformer encoder $\Upsilon$ acts on $Z$ and produces an output sequence $\Upsilon(Z)$.

To handle tasks that differ in data distributions, feature spaces, and label spaces, we employ components that (i) map inputs into a shared format for constructing the sequence $Z$, and (ii) project transformer outputs back into the original label space (see Fig. 1).

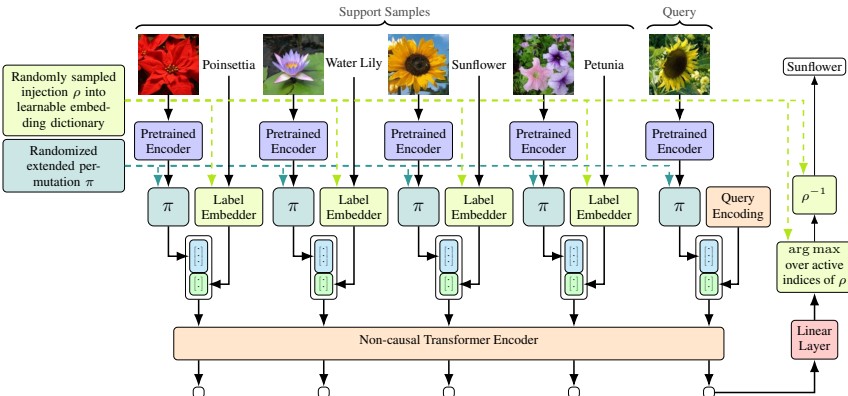

Figure 1: **Method overview**. The input is encoded with a modality-appropriate pretrained encoder and then projected to a common modality-agnostic space. The labels are embedded using a randomized injection to a learnable embedding dictionary. The input and label embeddings are concatenated and form the input tokens for a transformer encoder. A linear classification head makes a prediction in label embedding space, which is then remapped to the original set of labels.

## 3.1 Universal Feature Encoding

To handle varying feature domains $\mathcal{X}_T \neq \mathcal{X}_{T'}$ across tasks we encode the features from $\mathcal{X}_T$ into $\mathbb{R}^{d_T}$ with an encoder $\phi_T$ appropriate for the respective feature modality of each meta-training task. The encoder may take different forms, for example, it could be a simple concatenation of the input features or a pretrained feature extractor. We provide the encoder architecture details in Appendix D. The encoded vector is then projected into $\mathbb{R}^{d_{\text{data}}}$ with a randomly sampled projection $\pi : \mathbb{R}^{d_T} \to \mathbb{R}^{d_{\text{data}}}$. It makes sense to restrict this projection to preserve distances.

We sample $\pi$ uniformly at random from extended permutations (see Appendix A.4). This choice gives us two desirable properties. First, it avoids overfitting to the feature structure of a specific encoder $\phi_T$. Second, the random permutation of the feature space essentially acts as data augmentation, enabling our model to see more diverse inputs.

This is similar to how the human brain processes multi-modal information. The brain employs modality-specific processing in early sensory cortices that handle low-level features (Calvert, 2001). These specialized regions then feed into convergence zones, where information from different modalities is processed into increasingly abstract representations (Damasio, 1989). This mirrors our approach of using modality-specific encoders followed by a transformer which is shared across domains and reasons over the latent representations.

## 3.2 RANDOM INJECTION LABEL EMBEDDING AND CLASSIFICATION HEAD

We consider a general setting in which $\mathcal{Y}_T \not\cong \mathcal{Y}_{T'}$, i.e. the number of labels per task may vary. Importantly, the number of labels at test time can exceed the number seen during training. We refer to this as *label space extrapolation*. Since attention layers do not require architectural changes for longer sequences (more labels or more examples per label), the transformer scales naturally to other (larger) label spaces. We only need to design a label embedder and a classification head to be able to handle arbitrary $\mathcal{Y}_T$. To this end, we define a global learnable dictionary $\mathcal{E}$ of label embeddings $\{e_1, \ldots, e_M\} \subset \mathbb{R}^{d_{\text{label}}}$ with $M \gg 1$ and define $\mathcal{E}(i) = e_i$ for any index $i \in [M] = \{1, \ldots, M\}$. For any task for which $K := |\mathcal{Y}_T| \leq M$, we can simply use $K$ embeddings from the dictionary, thereby unifying the label space for all tasks. Crucially, if we make $M$ large enough, we can use this dictionary of embeddings for test tasks that have much larger labels sets than any of the training tasks. For each episode with task $T$ we sample an injective mapping $\rho : \mathcal{Y}_T \to [M]$ uniformly from the set of all injections $\text{Inj}(\mathcal{Y}_T, [M])$. The label embedder for this episode is now given $\mathcal{E} \circ \rho$, which maps elements of $\mathcal{Y}_T$ to vectors in the continuous space $\mathbb{R}^{d_{\text{label}}}$. We prove that this strategy meaningfully trains all embeddings even when $K \ll M$ for all tasks $T \in \mathcal{T}$ and moreover leads to label space extrapolation ability in Appendix A.

The last ingredient we need is a classifier head $\Psi$ acting on the transformer output $\Upsilon(Z)$. We use a linear layer $s$ to produce class scores for each index of the label embedding dictionary and compute $\hat{\jmath} = \arg\max_{j \in \rho(\mathcal{Y})} s_j$ where $\rho(\mathcal{Y}) \subset [M]$ is the image of the label space $\mathcal{Y}$ under $\rho$, restricting the possible indices to those of active embeddings in this episode. The classifier head is then given by $\Psi(\Lambda) = \rho^{-1}(\hat{\jmath})$ with $\hat{\jmath} = \arg\max_{j \in \rho(\mathcal{Y})}(s_j(\Lambda))$ and reverses the index selection in the embedder.

Input tokens are constructed by concatenating the feature encoding and label embedding. For the query token $z'$ we use a learnable query class marker $c \in \mathbb{R}^{d_{\text{label}}}$ in place of the label embedding.

## 3.3 TRAINING PROCEDURE

We train TAIL on a large-scale meta-dataset, consisting of ImageNet (Russakovsky et al., 2015), Meta-Album (Ullah et al., 2022) and MedIMeta (Woerner et al., 2025). We sample training episodes by first sampling a task from the meta-training set $\mathcal{T}_{\text{train}}$ and a "number of shots" $N$ for the episode and then sampling support and query sets as described in Section 2.5. The episode loss is given by the empirical risk $R_Q(f_{g_\theta}) = \sum_{(x,y) \in Q} \ell_T(g_\theta(S, x), y))$. Further details about the training procedure and chosen hyperparameters can be found in Appendix E.                                    R3.5

## 3.4 THEORETICAL PROPERTIES

As we established in Section 2.4 TAIL should be a valid learning algorithm for varying feature domains and label domains in order to satisfy the requirements of practical universality. We propose that the validity of the implicit algorithm learned by TAIL is invariant to both the feature domain and the label domain. This is ensured by the randomly sampled extended permutation and the random injection label embedding. We theoretically show the invariance in the feature domain by providing proofs for the coverage and unbiasedness of the embedding dictionary in Appendix A.               R2.2

Moreover, it is desirable that a learning algorithm's classification performance does not depend on the order of the demonstrations or on the identities of the labels, since the class labels in this setting are task-specific and only the relationship between the sample and the label is important, while the indexing of the labels is arbitrary. Therefore a learning algorithm's performance should be invariant to re-indexing of the labels and to the order in which the demonstrations are presented. TAIL fulfills both of these properties by being equivariant to label re-indexing and invariant to the demonstration order. We provide proofs for these propositions in Appendix A.

## 4 RELATED WORK

The majority of existing meta-learning approaches can be categorized as model-based, optimization-based, or metric-based. Model-based methods such as MANN (Santoro et al., 2016) learn a built-in learning algorithm which adapts to new tasks by changing its internal state. Optimization-based methods, such as Model-Agnostic Meta-Learning (MAML) (Finn et al., 2017), learn an initialization

that can be quickly adapted to new tasks with a few gradient updates. Metric-based methods, such as Prototypical Networks (Snell et al., 2017), learn a metric space where samples from the same class are close together. All of these approaches either cannot process varying feature spaces and label spaces, or their performance drops substantially when the evaluation domain is significantly different from the meta-training set (Chen et al., 2020; Luo et al., 2023).

It has recently been shown that the naive approach of fine-tuning or linear probing of large pre-trained models on a few-shot support set mostly outperforms meta-learning approaches (Guo et al., 2020; Oh et al., 2022). Moreover, fine-tuning or linear probing can be applied to different label spaces trivially by instantiating a new classification head. However, fine-tuning is computationally expensive when new tasks have to be learned frequently, since several gradient descent steps are required. Furthermore, weights for each task need to be stored in order to reuse the model at a later time for the same task. A simple alternative to fine-tuning is applying a ProtoNet head to a fixed pretrained backbone which allows meta-testing without any training at test time (see e.g. Fifty et al. (2023)). However, this approach heavily relies on the quality of the backbone. Prompt-based, adapter-based, or external-knowledge methods combining multiple foundation models can yield strong performance across domains, but often at the cost of requiring careful prompt design, fine-tuning, or reliance on pre-training domain coverage (Liu et al., 2024).  **R1.5**

Some early model-based meta-learning approaches (Santoro et al., 2016; Kaiser et al., 2017) reformulate few-shot classification as a sequence modeling problem using LSTMs (Hochreiter & Schmidhuber, 1997), which allows to solve unseen few-shot learning problems on the fly without retraining. More recent approaches pursue a similar strategy using self-attention sequence models (Vaswani et al., 2017). Mishra et al. (2018) apply temporal convolutions alternating with causal attention to the concatenated support and query data. Their model (SNAIL) treats the support set as a sequence of concatenated (input, label) pairs and directly produces query predictions. Due to its architecture requiring a fixed size for the features and labels, SNAIL cannot generalize across modalities or label sets. Newer works use transformers as few-shot learners, drawing parallels to in-context learning in NLP (Brown et al., 2020). GPICL (Kirsch et al., 2024) pursues a similar strategy using a causal transformer model, however, due to the use of positional encodings, the method can only process fixed-size sequences and can therefore not perform label space extrapolation. Because it uses random projections of the input for data augmentation the architecture is, in theory, suitable for processing different modalities.

The causal nature of the above approaches breaks the equivariance to label re-indexing and the demonstration order invariance (see Section 3.4). CAML by Fifty et al. (2023) instead uses a non-causal transformer and a Equal Length and Maximally Equiangular Set (ELMES) of vectors for embedding labels. Although ELMES vectors are fixed, the transformer needs to be trained to recognize them with $K$-way tasks to handle $K$-way tasks at test time and therefore CAML can also not perform label space extrapolation. Moreover, due to a fixed token size, CAML cannot work in the cross-modality setting.

Recent work compares in-context learning with meta-learning and studies how transformers learn in context. Oswald et al. (2023) show that transformers can approximate gradient descent in their forward pass. Bai et al. (2023) prove that transformers can implement a broad set of classical algorithms and perform in-context algorithm selection. Wu et al. (2025) view ICL models as meta-learners, arguing that transformers learn data-dependent learning algorithms by pretraining. We use this perspective to build a practical algorithm-implicit meta-learner that handles heterogeneous modalities and label spaces, and by providing the theoretical notion of practical universality. Multiple surveys  **R4.2**
(Hospedales et al., 2021; Vettoruzzo et al., 2024) present theoretical frameworks for meta-learning. In contrast to them, we expand the current taxonomy to surface the structural limitations preventing existing meta-learning methods from generalizing across domains and modalities.  **R3.1**

## 5 EXPERIMENTS

We evaluate TAIL across four different settings: performance on tasks with a similar domain (i.e. in-domain), cross-domain performance, generalization to unseen modalities, and label space extrapolation. For each test task and each $N$ we sample 1000 episodes with different support and query sets as described in Section 2.5. We report the mean accuracy over these 1000 episodes and the 95%-CI for the mean. Moreover, we assess the computational efficiency compared to the baselines. Our

Table 2: Mean classification accuracy in % over 1000 test episodes with 95% confidence interval.

| | 5-shot | | | | 1-shot | | | |
|---|---|---|---|---|---|---|---|---|
| | CIFAR-FS | *mini*ImageNet | *tiered*ImageNet | Pascal VOC | CIFAR-FS | *mini*ImageNet | *tiered*ImageNet | Pascal VOC |
| **Linear Probe** | 91.86 ± 0.32 | 97.65 ± 0.15 | 95.30 ± 0.30 | 83.57 ± 0.53 | 75.95 ± 0.64 | 88.22 ± 0.49 | 85.04 ± 0.61 | 64.22 ± 0.68 |
| **ProtoHead** | 91.09 ± 0.34 | 97.58 ± 0.15 | 95.54 ± 0.27 | 84.26 ± 0.51 | 73.24 ± 0.68 | 87.84 ± 0.51 | 83.76 ± 0.63 | 62.84 ± 0.77 |
| **SNAIL** | 91.03 ± 0.38 | 98.93 ± 0.11 | 97.43 ± 0.23 | 85.47 ± 0.53 | 76.37 ± 0.65 | 95.79 ± 0.30 | 92.35 ± 0.48 | 72.36 ± 0.75 |
| **GPICL** | 91.20 ± 0.35 | 99.44 ± 0.07 | 98.18 ± 0.19 | 87.46 ± 0.51 | 77.54 ± 0.66 | 97.64 ± 0.24 | 94.80 ± 0.41 | 75.61 ± 0.07 |
| **CAML** | 91.69 ± 0.34 | 99.29 ± 0.08 | 97.98 ± 0.21 | 87.87 ± 0.51 | 77.93 ± 0.65 | 97.08 ± 0.24 | 93.66 ± 0.42 | 76.76 ± 0.72 |
| **TAIL (ours)** | **94.55 ± 0.29** | **99.63 ± 0.06** | **98.67 ± 0.16** | **89.78 ± 0.48** | **84.35 ± 0.58** | **98.79 ± 0.14** | **96.76 ± 0.30** | **80.48 ± 0.73** |

Table 3: Mean 5-way 5-shot classification accuracy in % over 1000 test episodes

| | Aircraft* | CUB* | meta-iNat* | tiered meta-iNat* | cxr | oct | pbc | Paintings* | Pascal-Paintings* |
|---|---|---|---|---|---|---|---|---|---|
| **Linear Probe** | 92.12 ± 0.42 | 95.17 ± 0.30 | 92.91 ± 0.36 | 88.32 ± 0.47 | **25.10 ± 0.40** | 49.61 ± 0.55 | 68.39 ± 0.54 | **66.21 ± 0.44** | 71.62 ± 0.43 |
| **ProtoHead** | 92.07 ± 0.42 | 95.26 ± 0.30 | 92.51 ± 0.39 | 88.47 ± 0.44 | 25.04 ± 0.41 | 49.37 ± 0.53 | 69.81 ± 0.51 | 66.06 ± 0.46 | 71.57 ± 0.44 |
| **SNAIL** | 90.19 ± 0.48 | 95.57 ± 0.34 | 95.08 ± 0.33 | 91.21 ± 0.47 | 21.52 ± 0.30 | 35.36 ± 0.46 | 71.08 ± 0.54 | 57.49 ± 0.46 | 69.12 ± 0.46 |
| **GPICL** | 90.83 ± 0.45 | 96.02 ± 0.30 | 95.56 ± 0.31 | 91.15 ± 0.46 | 22.70 ± 0.34 | 41.64 ± 0.05 | 53.28 ± 0.52 | 54.29 ± 0.48 | 65.79 ± 0.51 |
| **CAML** | 93.09 ± 0.39 | 97.55 ± 0.22 | 96.20 ± 0.28 | 93.53 ± 0.36 | 23.30 ± 0.40 | 46.48 ± 0.53 | 80.19 ± 0.45 | 58.72 ± 0.49 | 70.13 ± 0.46 |
| **TAIL (ours)** | **95.01 ± 0.34** | **98.51 ± 0.17** | **97.70 ± 0.21** | **95.69 ± 0.31** | 23.68 ± 0.38 | **50.64 ± 0.52** | **84.96 ± 0.42** | 63.03 ± 0.48 | **73.04 ± 0.47** |

experiments show that TAIL achieves state-of-the-art results while having the flexibility to handle completely new domains and task configurations without retraining. We additionally report ablation studies to analyze the effect of the universal feature encoding using random projections, and the embedding schedule of the embedding dictionary. The results are reported in Appendix C.

## 5.1 BASELINES AND TRAINING

We consider two algorithm-explicit, and three algorithm-implicit baselines. First, we consider the algorithm-explicit **Linear Probing** approach, which trains a linear classifier on representations from pretrained foundation models at meta-test time. This can be considered a universal learning algorithm and has been shown to perform well on specialized datasets (Woerner & Baumgartner, 2024), but requires retraining at test time for every task. We further consider the algorithm-explicit ProtoNet (Snell et al., 2017) with a fixed pre-trained backbone (which we coin **ProtoHead**), which is not meta-trained, but allows meta-testing without retraining at test time. Lastly, we consdier the attention-based algorithm-implicit approaches **SNAIL** (Mishra et al., 2018), **CAML** (Fifty et al., 2023) and **GPICL** (Kirsch et al., 2024) which are closest to our own approach. We make slight modifications to GPICL, to able to process different label spaces (see Appendix D.3). For all baseline methods, as well as TAIL (ours), we use fixed pretrained backbones as feature encoders. For a fair comparison and to ensure that TAIL does not gain an advantage over the baselines solely due to its meta-training set, we meta-trained all meta-learning baselines on the same meta-training set as TAIL. Moreover, the same pretrained encoders are used for each of the baselines. All subsequent experiments are performed on these models.

**R1.1**

## 5.2 RESULTS

**In-Domain Image Classification:** In order to evaluate the in-domain performance on standard benchmark problems, we tested all approaches on MiniImageNet (Vinyals et al., 2016), tieredImageNet (Ren et al., 2018), CIFAR-FS (Bertinetto et al., 2018), and Pascal VOC (Everingham et al., 2010), which are generic object-recognition datasets and therefore can be considered in-domain with respect to our training set. Note that in-domain refers to test tasks that are still unseen at training time but whose image distributions are similar to those in the training set. As shown in Table 2, TAIL consistently outperformed all baselines in the 1-shot and 5-shot setting.

**Cross-Domain Specialized Datasets:** For the cross-domain evaluation, we tested all learners on a diverse set of specialized domains not present in our meta-training set, including medical imaging tasks, aircraft recognition, and artistic domains. We used the Caltech Birds Dataset (CUB) (Wah et al., 2011), FGVC-Aircraft (Maji et al., 2013), meta-iNat and tiered meta-iNat (Wertheimer & Hariharan, 2019), the cxr, oct and pbc subsets from MedIMeta (Woerner et al., 2025), the Paintings dataset (Crowley & Zisserman, 2015) and the Inter-Domain Image Classification Pascal+Paintings (Fifty et al., 2023) dataset.

The 5-shot, and 1-shot performance in the out-of-domain setting are shown in Tables 3 and4, respectively. Some dataset may contain certain classes that have a semantic overlap with our training set. We have marked these with an asterisk. We are certain that the medical datasets from MedImeta

Table 4: Mean 5-way 1-shot classification accuracy in % over 1000 test episodes.

| | Aircraft* | CUB* | meta-iNat* | tiered meta-iNat* | cxr | oct | pbc | Paintings* | Pascal-Paintings* |
|---|---|---|---|---|---|---|---|---|---|
| **Linear Probe** | 80.32 ± 0.70 | 83.06 ± 0.66 | 79.02 ± 0.68 | 71.91 ± 0.72 | **22.35 ± 0.35** | 36.29 ± 0.53 | 45.91 ± 0.59 | **49.96 ± 0.56** | 51.14 ± 0.56 |
| **ProtoHead** | 78.89 ± 0.72 | 81.95 ± 0.71 | 78.31 ± 0.69 | 70.48 ± 0.75 | 22.33 ± 0.36 | 36.25 ± 0.52 | 45.29 ± 0.58 | 48.64 ± 0.58 | 50.58 ± 0.54 |
| **SNAIL** | 80.82 ± 0.07 | 88.96 ± 0.58 | 87.30 ± 0.60 | 81.81 ± 0.68 | 20.61 ± 0.30 | 29.93 ± 0.45 | 60.26 ± 0.67 | 45.01 ± 0.57 | 51.82 ± 0.61 |
| **GPICL** | 74.30 ± 0.80 | 85.63 ± 0.65 | 88.92 ± 0.57 | 78.29 ± 0.74 | 20.72 ± 0.29 | 30.71 ± 0.44 | 30.63 ± 0.48 | 41.63 ± 0.56 | 49.91 ± 0.57 |
| **CAML** | 84.33 ± 0.65 | 92.34 ± 0.49 | 90.74 ± 0.52 | 84.28 ± 0.64 | 21.86 ± 0.36 | 35.53 ± 0.54 | 61.73 ± 0.67 | 45.77 ± 0.06 | 53.28 ± 0.63 |
| **TAIL (ours)** | **89.42 ± 0.56** | **95.51 ± 0.38** | **93.84 ± 0.42** | **90.23 ± 0.53** | 22.15 ± 0.38 | **36.74 ± 0.58** | **70.25 ± 0.66** | 48.00 ± 0.61 | **55.71 ± 0.63** |

Table 5: Cross-modal performance: models were trained on image classification tasks and tested on on text classification tasks.

| | 5-shot | 1-shot |
|---|---|---|
| Linear Probe | 89.33 ± 0.44 | **85.32 ± 1.03** |
| ProtoHead | 88.92 ± 0.44 | 83.86 ± 1.13 |
| GPICL (trained on images) | 50.88 ± 0.42 | 50.31 ± 0.28 |
| TAIL (ours) (trained on images) | **89.62 ± 0.48** | 84.87 ± 1.04 |

do not contain classes with a semantic overlap. TAIL achieved state-of-the-art performance on the majority of datasets and is competitive on the two remaining datasets, without any domain-specific retraining. We note that the accuracy for all methods is lower in the more challenging 1-shot setting. **R3.4**

**Cross-Modal Generalization to Unseen Modalities:** The key test of practical universality is the ability to generalize to completely different modalities without retraining. To assess the limits of generalizability, we evaluated the models trained exclusively on images on a text classification problem. Here, we only included the baselines which architecturally permit operating in a different feature space than was used in the meta-training stage: Linear Probing, ProtoHead, GPICL and TAIL. We used sentiment classification of IMDB movie reviews (Maas et al., 2011) as a the text classification task for our evaluation. The results in Table 5 show that our model achieves superior cross-modal generalization compared to all approaches in the 5-shot setting, and is only slightly outperformed by the naive linear probing approach in the 1-shot setting. Out of the meta-learned methods, TAIL maintains the strongest performance when applied to completely different modalities. While GPICL can theoretically handle different modalities, its performance degrades severely, to the point that the accuracy is on the level of random chance. We note that CAML and SNAIL cannot process features from domains with different dimensionality than its training domain.

**Label Space Extrapolation:** Traditional meta-learning methods fail when confronted with tasks containing more classes than seen during training. We demonstrate that TAIL gracefully handles label space extrapolation, maintaining reasonable performance even with $20\times$ more classes than during meta-training. To illustrate this, we used the meta-trained learners from Section 5.2, which were trained only on task instances with $K \leq 5$. We then evaluated performance as the number of classes increases up to 100-way classification. We additionally took advantage of TAIL's computational efficiency to train a version of TAIL with 50 labels used in the meta-training stage (TAIL 50w), which is computationally infeasible for the other attention-based approaches.

As can be seen in Figure 2 (left), performance degraded with more labels $K$ per tasks as is expected. TAIL achieved the top performance until 70-way classification tasks, where it was outperformed by Linear Probing and ProtoHead which require a domain-specific classification head. We further note that TAIL trained with tasks up to 50 labels significantly outperforms the baselines throughout the testing scenario.

**Computational Efficiency for Large Label Sets:** While GPICL (Kirsch et al., 2024) and CAML (Fifty et al., 2023) can theoretically handle arbitrary label spaces, in practice those methods are severely computationally limited at larger task sizes. While evaluation with large label spaces is still possible up to a point, training with label spaces larger than 20 is computationally prohibitive on current computational infrastructures.

In contrast, TAIL provides dramatic computational advantages over existing attention-based meta-learning approaches. To illustrate this, we measured the wall clock time for solving a 1-shot meta-test task with increasing numbers of labels ($K$). As can be seen in Figure 2 (c), the wall clock time of GPICL and CAML increases very rapidly with increasing $K$, while TAIL retains a similar computational complexity to the methods not based on transformers (i.e. Linear Probing and ProtoHead). Training time and memory usage (Figure 2 (d,e)) show an even more dramatic difference. Training GPICL and CAML becomes infeasible for $K \geq 20$. Figure 2 (b) shows that TAIL in fact faster than Linear Probing for typical task sizes below $K = 70$ since it does not require training at meta-test

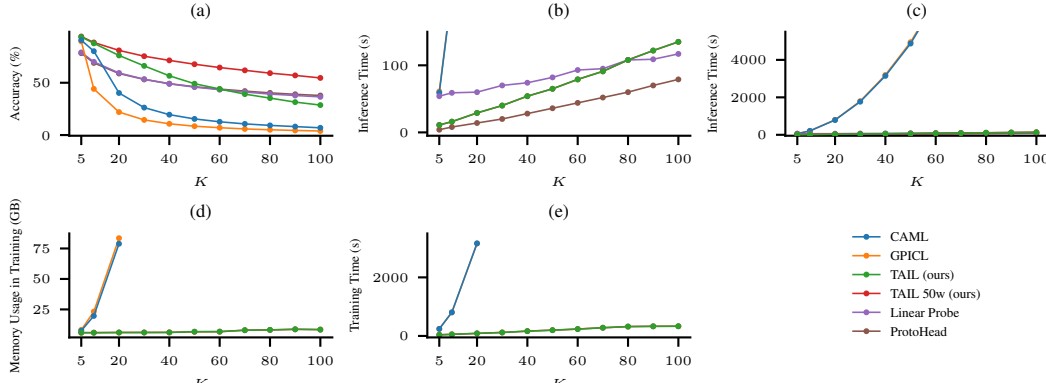

Figure 2: **(a)**: performance degradation with increasing number of classes (1-shot setting). **(b)** and **(c)**: wall clock time for 1000 test episodes as a function of task size. Two different scales show the relation to the algorithm-explicit baselines and to the meta-learning baselines. **(d)**: memory usage during training as a function of task size, **(c)**: wall clock time for 1000 training episodes.       R3.6
R1.2, R3.3, R3.6

time and avoids the per-task optimization loop. The results for the 5-way task look similar and are       R3.6
reported in Fig. 3 in the Appendix. We measured only the time and memory requirements of TAIL
and the baselines without the predtrained encoder.       R1.2, R3.3

## 5.3 INSIGHTS AND DISCUSSION

**The Power of Algorithm-Implicit Learning.** Our results confirm that allowing the learning algorithm to emerge from data, rather than being explicitly specified, provides fundamental advantages in few-shot settings. TAIL mostly outperforms methods with fixed algorithmic assumptions. Simple algorithm-explicit methods, such as gradient descent, still likely outperform universal meta-learners in settings with limited meta-training data due to their strong inductive biases. However, we showed that TAIL successfully transfers to new modalities, meaning that TAIL can learn from a broad array of data without being restricted to in-domain tasks and therefore is a powerful option for many tasks.       R2.5

**Label Space Extrapolation Capability.** The ability to handle tasks with up to $20\times$ more classes than seen during training (Figure 2) is a qualitative leap in meta-learning.

**Cross-Modal Transfer Without Retraining.** The successful application to text classification (Table 5) without any architectural modifications or retraining validates our universal feature encoding approach. This result has significant practical implications, as it eliminates the need for modality-specific meta-training datasets.

**Computational Efficiency at Scale.** The substantial computational advantages over CAML and GPICL (Figure 2, right panel) stem from our architectural choices. By processing all query samples jointly and using a single forward pass through the transformer, we achieve orders of magnitude speedup. This efficiency gap widens as task complexity increases.

## 6 CONCLUSION

We introduced a theoretical framework for meta-learning and TAIL, a meta-learning approach that achieves practical universality in few-shot learning. Our contributions include random projections for cross-modal generalization and label embeddings with a global dictionary to scale to far more classes than seen in training. Empirically, TAIL sets new state-of-the-art results, generalizes to unseen modalities (e.g., 89.6% on text after training only on images), handles tasks with up to 100 classes, and offers large computational savings.

Our theoretical framework introduces a distinction between *algorithm-explicit* and *algorithm-implicit* learning systems and the notion of practical universality.

Directions for future work include extending the approach to other learning scenarios beyond classification, such as regression and structured prediction, and investigating whether similar principles of algorithm-implicit learning are also superior in other settings like reinforcement learning.

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

## A THEORETICAL ANALYSIS AND PROOFS

In this Appendix, we formalize and prove the theoretical properties described in Section **??**.

### A.1 PERMUTATION EQUIVARIANCE WITH RESPECT TO THE LABEL SPACE

**Theorem 1** (Equivariance to label re-indexing). *Let $\mathcal{X}$ be a feature space, $\mathcal{Y}$ a label space and $S = \{(x_i, y_i)\}_{i=1}^{n}$ a support dataset and $(x, y)$ a query sample. For any permutation $\sigma$ of $\mathcal{Y}$, let*

$$S^\sigma = \{(x_i, \sigma(y_i))\}_i, \quad y_{\mathrm{q}}^\sigma = \sigma(y_{\mathrm{q}}).$$

*Then*

$$g(S^\sigma, x) \stackrel{d}{=} \sigma(g(S, x)).$$

*i.e. $g_\theta(S, x)$ is equivariant in distribution to the reindexing of $\mathcal{Y}$.*

*Proof of Theorem 1.* Let $\mathcal{E}$ be our embedding dictionary and $\rho : \mathcal{Y} \to [M]$ be an injection sampled uniformly from the set of all injections $\mathrm{Inj}(\mathcal{Y}_T, [M])$. Define $\rho' := \rho \circ \sigma^{-1}$. We note that the images under $\rho$ and $\rho'$ are equal, i.e. $\rho(\mathcal{Y}) = \rho'(\mathcal{Y})$. Then

$$g(S^\sigma, x\,; \rho') = \rho'^{-1}\left( \underset{j \in \rho'(\mathcal{Y})}{\arg\max}\ s_j \left( \Upsilon\left( \begin{bmatrix} \pi(\phi(x_1)) & \dots & \pi(\phi(x_n)) & \pi(\phi(x)) \\ \mathcal{E}(\rho'(\sigma(y_1))) & \dots & \mathcal{E}(\rho'(\sigma(y_n))) & c \end{bmatrix} \right) \right) \right)$$

$$= \sigma\left( \rho^{-1}\left( \underset{j \in \rho(\mathcal{Y})}{\arg\max}\ s_j \left( \Upsilon\left( \begin{bmatrix} \pi(\phi(x_1)) & \dots & \pi(\phi(x_n)) & \pi(\phi(x)) \\ \mathcal{E}(\rho(y_1)) & \dots & \mathcal{E}(\rho(y_n)) & c \end{bmatrix} \right) \right) \right) \right) = \sigma\left( g(S, x\,; \rho) \right).$$

Since $\rho$ is sampled uniformly over injections, $\rho' := \rho \circ \sigma^{-1}$ and $\rho$ have the same probability. Combining, the prediction distributions are identical. $\qquad\square$

### A.2 COVERAGE OF THE EMBEDDING DICTIONARY

Even though each episode only uses $K$ out of $M$ embeddings, we show that across episodes all embeddings and their corresponding "detectors" in the transformer are trained.

**Proposition 1** (Unbiased gradients). *For each embedding $e_j$, the episode gradient satisfies*

$$\underset{\rho}{\mathbb{E}}[\nabla_{e_j} \ell] = \tfrac{K}{M}\ \mathbb{E}\big[\nabla_{e_j} \ell \mid j \in S\big].$$

*Thus stochastic gradients are unbiased up to the constant factor $\frac{K}{M}$.*

**Proposition 2** (Coverage over $t$ episodes). *Let $N_j(t)$ be the number of episodes in which $e_j$ is included. Then*

$$N_j(t) \sim \mathrm{Binomial}\big(t, \tfrac{K}{M}\big), \quad \mathbb{E}[N_j(t)] = \tfrac{tK}{M}.$$

*By Chernoff bound, for any $\delta \in (0, 1)$,*

$$\Pr\big[N_j(t) \le (1 - \delta)\tfrac{tK}{M}\big] \le \exp\left(-\frac{\delta^2 tK}{2M}\right).$$

*It follows that with high probability every embedding is updated $\Omega(\frac{tK}{M})$ times once $t \gtrsim \frac{M}{K}\log M$.*

*Remark A.1.* Since every episode includes at least one label embedding, transformer parameters $\theta_\Upsilon$ interacting with embeddings receive gradient updates in every episode. By symmetry of $\rho$, these "detectors" are trained uniformly across all embeddings.

### A.3 DEMONSTRATION ORDER INVARIANCE

It is a desirable quality for the DCI predictor to be invariant to the order in which the support set is presented.

**Theorem 2** (Demonstration Order Invariance). *Let $S$ be a sequence of support samples $S = ((x_i, y_i))_{i=1}^{n}$ and $(x, y)$ a query sample. For any permutation $\sigma$ of $S$*

$$g(\sigma(S), x) = g(S, x).$$

*i.e. $g_\theta(S, x)$ is invariant to the order of demonstrations.*

*Proof.* Building on (Kossen et al., 2021, Appendix A), we only need to prove that the domain-specific encoder and the random injection embedding are equivariant to the order of demonstrations. Since $\phi, \rho$ are elementwise applied the input sequence and do not depend on position, it trivially follows that embedding the sequence is permutation equivariant. Since the transformer $\Upsilon$ itself is permutation invariant following (Kossen et al., 2021, Appendix A), and $\Psi$ only operates on the index of the query sample, $g_\theta$ is invariant to permutations. $\qquad\square$

### A.4 EXTENDED PERMUTATIONS

**Definition 7** (Extended permutation). Let $n \leq k$. An *extended permutation matrix* is a binary matrix $E \in \{0,1\}^{k \times n}$ such that each column contains exactly one 1, each row contains at most one 1. Equivalently, $E$ encodes an injective map $\pi : \{1, \ldots, n\} \rightarrow \{1, \ldots, k\}$.

## B 5-SHOT ACCURACY DEGRADATION

We additionally report the accuracy degradation on 5-shot tasks in figure 3.

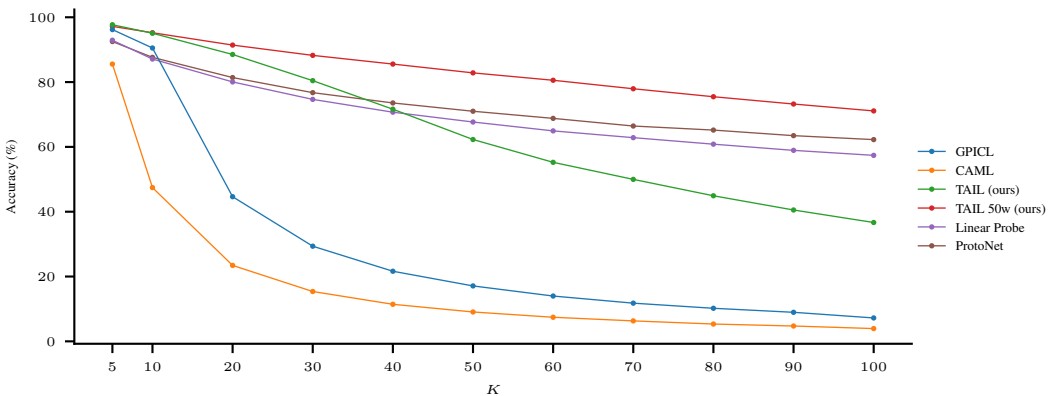

Figure 3: Performance degradation with increasing number of classes (5-shot setting).

## C ABLATION STUDIES

In order to better understand our method's reliance of our key architectural components, we conducted the following ablation experiments.

**Random feature projection:** First we compared TAIL, to a TAIL with random projections without the restriction to extended permutations and to a TAIL *without* the universal feature encoding using random projections of the feature space described in Section. 3.1. As can be seen in Table 6, TAIL's performance decreased without the random extended permutation, especially in the cross-domain and cross-modality settings. This indicates that the random projection mechanism is necessary for generalization to unseen domains.

Table 6: Average performance on in-domain datasets, cross-domain datasets and cross-modality datasets with and without the random permutation into a common latent space.

| | 5-shot | | | 1-shot | | |
|---|---|---|---|---|---|---|
| | in-domain | cross-domain | cross-modality | in-domain | cross-domain | cross-modality |
| TAIL *without* random $\pi$ | $98.75 \pm 0.06$ | $83.07 \pm 0.11$ | $87.43 \pm 0.62$ | $95.80 \pm 0.16$ | $63.94 \pm 0.17$ | $66.69 \pm 1.62$ |
| TAIL, random projection $\pi$ | $99.09 \pm 0.07$ | $86.83 \pm 0.10$ | $89.91 \pm 0.44$ | $96.60 \pm 0.13$ | $67.70 \pm 0.16$ | $84.96 \pm 0.98$ |
| **TAIL, random extended permutation $\pi$** | $99.21 \pm 0.06$ | $87.58 \pm 0.10$ | $89.62 \pm 0.48$ | $97.30 \pm 0.12$ | $67.80 \pm 0.17$ | $84.87 \pm 1.04$ |

**R1.3, R2.4, R4.4**

**Causal vs. non-causal architecture:** To quantify the impact of removing the causal mask, we replace TAIL's non-causal transformer with an architecture identical in all respects except for a

standard causal attention mask. Across all settings, the causal variant exhibits a consistent and substantial drop in accuracy.

R1.4, R2.4, R4.4

Table 7: Average performance on in-domain datasets, cross-domain datasets and cross-modality datasets with a causal and with a non-causal transformer architecture.

|  | 5-shot | | | 1-shot | | |
|---|---|---|---|---|---|---|
|  | in-domain | cross-domain | cross-modality | in-domain | cross-domain | cross-modality |
| TAIL with causal architecture | 98.81 ± 0.06 | 70.60 ± 0.13 | 70.08 ± 0.71 | 93.67 ± 0.19 | 53.74 ± 0.17 | 66.91 ± 1.42 |
| **TAIL (non causal)** | 99.21 ± 0.06 | 87.58 ± 0.10 | 89.62 ± 0.48 | 97.30 ± 0.12 | 67.80 ± 0.17 | 84.87 ± 1.04 |

R1.4, R2.4, R4.4

**Mixed-modality training:** We explored the impact of adding text data to the meta-training set. As table 8 shows, the mixed-modality training does not significantly improve performance, except for the text classification task itself. We believe this is an artifact of the composition of the meta-dataset: the comparatively small amount of text data does not significantly increase the data diversity provided by our large image-classification meta-training set. We speculate that exposing TAIL to a larger cross-modal meta-training set with a broader variety of feature domains could strengthen the learned algorithm and add robustness on cross-modality evaluation. The very minor improvement on the text-classification task supports the claim the TAIL generalizes well to unseen modalities.

R1.6

Table 8: Average performance on in-domain datasets, cross-domain datasets and cross-modality datasets with our default meta-training set and with a mixed-modality meta-training set including text-classifcation tasks.

|  | 5-shot | | | 1-shot | | |
|---|---|---|---|---|---|---|
|  | in-domain | cross-domain | cross-modality | in-domain | cross-domain | cross-modality |
| TAIL trained on mixed-modality training set | 99.06 ± 0.06 | 85.19 ± 0.09 | 90.10 ± 0.59 | 96.75 ± 0.15 | 68.91 ± 0.20 | 85.91 ± 1.03 |
| **TAIL (trained on images only)** | 99.21 ± 0.06 | 87.58 ± 0.10 | 89.62 ± 0.48 | 97.30 ± 0.12 | 67.80 ± 0.17 | 84.87 ± 1.04 |

R1.6

**Training schedule for the label embedding dictionary:** Lastly, we investigated the effect the embedding dictionary schedule (see Section 3.2 on the speed of convergence. Figure 4 shows that slowly adding more embeddings to the embedding dictionary during training accelerates convergence. This is likely due to the fact that training can be jump-started with easier problems, an effect that is also known from curriculum learning (Bengio et al., 2009).

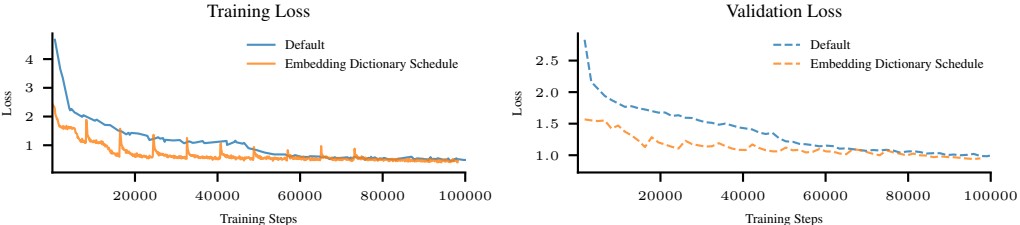

Figure 4: Validation loss curves for scheduled addition of more embeddings to the embedding dictionary.

# D ARCHITECTURE DETAILS

## D.1 PRETRAINED FEATURE ENCODERS

We use the following pretrained encoders for the different modalities:

**Vision Tasks:** We use the ViT-H model trained on LAION-2B (Schuhmann et al., 2022) provided by OpenCLIP (Ilharco et al., 2021), an open source reimplementation of OpenAI's CLIP. Images were resized to $224 \times 224$ and we applied standard ImageNet normalization.

**Text Tasks:** We use a pretrained uncased version of DistilBERT (Sanh et al., 2020) to embed text tasks.

## D.2  TRANSFORMER ARCHITECTURE

Table 9: TAIL architecture hyperparameters

| Component | Configuration |
| --- | --- |
| **Transformer Encoder** | |
| Hidden dimension | 1536 |
| Number of layers | 16 |
| Attention heads | 16 |
| MLP dimension | 3072 |
| Dropout | 0.0 |
| Activation | GELU |
| Normalization | Layer Norm |
| **Feature Projection** | |
| Output dimension ($d_{\text{data}}$) | 1280 |
| Type | Extended permutation matrix |
| **Label Embedding** | |
| Embedding dimension ($d_{\text{label}}$) | 256 |
| Dictionary size ($M$) | 100 (default), 256 (large) |

## D.3  MODIFICATIONS TO GPICL

We extend the positional encoding to have more vectors than required and use the first $K \cdot N$ vectors to encode the positions of a sequence. With this approach, GPICL accepts sequences of variable lengths, allowing us to test it in the cross-modality and label extrapolation settings.

# E  TRAINING DETAILS

## E.1  EPISODE SAMPLING

For each episode, a dataset is sampled at random from the meta-training set, weighted by the number of classes in the dataset. A task is generated by choosing $K$ classes at random from the dataset. Suppoort and Query sets are then generated by first sampling a "number of shots" $N$ for the episode and then sampling support and query sets as described in Section 2.5.

## E.2  OPTIMIZATION

We use the Adam optimizer with a circular learning rate schedule and a maximum learning rate of $3 \cdot 10^{-5}$.

**R3.5**

## DISCLAIMER FOR USE OF LLMS

We primarily used LLMs in coding co-pilot applications to facilitate experimentation and help with plotting code for result presentation. LLMs were also used as writing tools to assist in refining the paper. However, the final version was carefully reviewed and finalized by the authors. No LLMs were used in ideation and experimental design.

