# OpenReview forum: "Universal Algorithm-Implicit Learning"
_ICLR.cc/2026/Conference — Submitted to ICLR 2026_

### Official Review · Reviewer_RReC · 2025-10-15

**Soundness:** 3
**Presentation:** 2
**Contribution:** 1
**Rating:** 2
**Confidence:** 4

**Summary:**

This paper presents TAIL (Transformer-based Algorithm-Implicit Learner), a meta-learner designed for practical universality. The key idea is to reframe few-shot learning as a sequence modeling problem, where a non-causal transformer learns an implicit learning algorithm. The authors introduce two main technical contributions to handle diverse tasks: a universal feature encoding scheme using random projections into a common latent space, and a universal label handling mechanism using a randomized global dictionary of learnable embeddings. The paper also provides a theoretical framework that distinguishes between algorithm-explicit and algorithm-implicit learning. The experimental results demonstrate that TAIL achieves state-of-the-art performance on a variety of benchmarks,

**Strengths:**

This paper gives a clear and  general definition and discussion about learning problems and learning to learn, which helps readers understand.

The paper presents a comprehensive set of experiments that demonstrate the effectiveness of TAIL. The method is shown to outperform strong baselines across a wide range of settings, including in-domain, cross-domain, cross-modal, and label-space extrapolation scenarios.

**Weaknesses:**

1. Lack of novelty and contribution. The framework, and the claimed technical contribution of a universal feature encoding scheme and a universal label handling mechanism, and most of the theoretical proofs, are almost identical with [1] [2].

2. Lack of key references. A line of works, which discuss what learning algorithms does ICL model learn, are highly related [3][4]. Especially [5] has exactly pointed out that ICL models are meta-learners with minimal bias, and learning implicit learning algorithms.

3. The reviewer could not agree with the definition of algorithm-explicit/implicit learning in section 2.3. The paper defines the distinction as if its training procedure is explicitly specified, and specify GD as explicitly specified while attention-based meta-learners are not. However, what is "explicitly specified" is not clear. For example, one could specify the training process as the forward-propagation process in transformer with certain training set and query as sequence input, in which case attention-based meta-learners are algorithm-explicit learning.

4.Lack of in-depth analysis of results and ablation study. Considering the similarity among the TAIL and [1][2], the reviewer could not understand why TAIL outperforms the others.

5. Obvious typos, even at the beginning of abstract.

[1] General-purpose in-context learning by meta-learning transformers
[2] Context-aware meta-learning, ICLR 2024
[3] Transformers Learn In-Context by Gradient Descent, ICML 2023
[4] Transformers as Statisticians: Provable In-Context Learning with In-Context Algorithm Selection, NIPS 2023
[5] Why In-Context Learning Models are Good Few-Shot Learners?, ICLR2025

**Questions:**

Please refer to weakness

---

> ### Author Response · Authors · 2025-11-21
> **Response to Reviewer RReC (R4)**
>
> Thank you for the detailed and thoughtful review, and for recognizing the clarity of our theoretical framework and the breadth of our experimental results. In the following, we address your comments and questions in detail.
>
>
> ---
>
>
> ### **R4.1. Novelty and similarity to [1] and [2]**
>
>
> > *“The framework, and the claimed technical contribution of a universal feature encoding scheme and a universal label handling mechanism, and most of the theoretical proofs, are almost identical with [1] [2].”*
>
>
> Thank you for raising this point but we respectfully disagree that TAIL is “almost identical’’ to these prior works. As clarified in our Related Work section, earlier model-based meta-learners, including SNAIL, CAML and GPICL, reformulate few-shot classification as a sequence modeling problem, but **none** of these methods can handle both heterogeneous feature spaces and arbitrary label spaces. Their architectures require fixed feature dimensions, fixed label embeddings, or fixed sequence lengths, which prevents cross-modality generalization and label-space extrapolation.
>
>
> To make these distinctions explicit, we provide the following comparison table:
>
>
> | **Method**                     | **Causal / Non-causal** | **Supports Variable Feature Spaces (Cross-Modality)** | **Supports Arbitrary Label Spaces (Label-Space Extrapolation)** | **Sequence Length Flexibility**                         | **Key Limitation**                                           |
> | ------------------------------ | ----------------------- | ----------------------------------------------------- | --------------------------------------------------------------- | ------------------------------------------------------- | ------------------------------------------------------------ |
> | **SNAIL** (Mishra et al. 2018) | Causal                  | ✗ fixed-dimensional concatenation                     | ✗ labels must be fixed-size embeddings                          | Limited (due to causal temporal conv)                   | Cannot generalize across modalities or label sets            |
> | **GPICL** (Kirsch et al. 2024) | Causal                  | (theoretically possible but not demonstrated)         | ✗ positional encodings tie model to fixed number of classes     | ✗ sequence length fixed by positional encodings         | No label extrapolation; no cross-modality experiments        |
> | **CAML** (Fifty et al. 2023)   | Non-causal              | ✗ fixed token size prevents modality transfer         | ✗ uses ELMES embeddings learned for specific (K)                | ✓ flexible                                              | Cannot extrapolate to unseen label spaces; modality-specific |
> | **TAIL (ours)**                | **Non-causal**          | **✓ via universal randomized projection**             | **✓ via universal label dictionary**                            | **✓ supports arbitrary number of support/query tokens** | —                                                            |
>
>
> Beyond architectural distinctions, we also want to clarify the point regarding the theoretical analysis. Our proofs establish **desirable properties specifically for TAIL’s architecture**, and several of our propositions have **no analogue** in the cited works. While different papers may show similar high-level properties for their own models, this does not imply that the underlying constructions or proofs are the same. Our theoretical analysis addresses questions that arise uniquely from TAIL’s universal-projection and label dictionary mechanisms.
>
>
> Our technical contributions allow us to work in a completely novel problem setting with arbitrary label cardinalities and modalities. In addition, our framework for **practical universality** provides a formal definition of the novel problem setting which to our knowledge we are the first to provide.
>
>
> ---
>
>
> ### **R4.2. Missing related work [3], [4], [5]**
>
>
> > *“A line of works, which discuss what learning algorithms does ICL model learn, are highly related.”*
>
>
> Thank you for pointing us to this line of work. We agree that these papers are highly relevant and complement our work. In the revised version, we have added a concise discussion to our Related Work section.
>
>
> ---

---

> ### Author Response · Authors · 2025-11-21
> **Response to Reviewer RReC (R4)**
>
> ### **R4.3. Ambiguity in “algorithm-explicit” vs. “algorithm-implicit”**
>
>
> > *“The definition of algorithm-explicit/implicit learning [and] what is 'explicitly specified' is not clear. For example, one could specify [… the transformer forward pass as an explicit algorithm…]”*
>
>
> We agree with the reviewer’s observation that our terminology can be perceived as ambiguous. Nevertheless, we believe that this is the definition that makes the clearest distinction between the two groups of algorithms.
> In internal discussions prior to the submission of this paper, we discussed instead making the distinction between **model-explicit** and **model-implicit**, i.e. whether a model/hypothesis is expressed which can be used for prediction or a model is only implicitly defined by the behaviour of the DCI function.
> While the concept is similar, we ultimately felt that **algorithm-implicit/explicit** was more precise, since we are focusing on the expressive power vs. the biases of the learning algorithm.
>
>
> Our intended distinction is:
> * **explicit** = the algorithm (or update rule) is *externally specified* (e.g., gradient descent),
> * **implicit** = the algorithm or DCI function is *entirely learned as part of the meta-learner’s forward computation*.
>
>
> We aim to clarify this boundary and revise wording to reduce ambiguity in next week’s revision.
> However, we are open to suggestions for a less ambiguous taxonomy.
>
>
> ---
>
>
> ### **R4.4. Why TAIL outperforms [1], [2]**
>
>
> > *“Lack of in-depth analysis of results and ablation study. Considering the similarity among the TAIL and [1][2], the reviewer could not understand why TAIL outperforms the others.”*
>
>
> Thank you for highlighting this. We agree that our paper will benefit from a more explicit analysis of why TAIL outperforms prior model-based meta-learners. Although the appendix already includes ablations, we recognize that the contribution of some architectural choices was not sufficiently isolated.
>
>
> However, we would like to emphasize that TAIL is **not** architecturally identical to [1] and [2]. We refer to the comparison table above. In particular:
>
>
> - GPICL [1] uses causal masking, which imposes an arbitrary ordering on demonstrations, and moreover does not represent the data and label of a particular demonstration in the same token.
> - TAIL’s ability to process heterogeneous feature spaces relies on the randomized projection. Prior works assume a fixed-dimensional feature space.
> - Unlike ELMES or one-hot label embeddings used in prior models, our embedding dictionary enables label-space extrapolation.
>
>
> To make the impact of each of these differences clearer, we will add additional ablation studies to include an experiment for each of these.
>
>
> We expect to complete these additional ablation experiments by next week.
>
>
> ---
>
>
> ### **R4.5. Typos**
>
>
> > *“Obvious typos, even at the beginning of abstract.”*
>
>
> Thank you for pointing this out! We have corrected the typo in the abstract, and will perform a thorough proof-reading of the whole manuscript.
>
>
> ---

---

> > ### Author Response · Authors · 2025-12-04
> > **Follow-ups for Reviewer RReC (R4)**
> >
> > ### R4.3 – Clarifying “algorithm-explicit” vs “algorithm-implicit”
> >
> >
> > We have revised Section 2 to clarify the terminology around **algorithm-explicit** vs **algorithm-implicit** learning systems, and to more clearly state the intended distinction (externally specified vs internally learned rules).
> >
> >
> > ---
> >
> >
> > ### R4.4 – Additional ablations explaining why TAIL outperforms [1,2]
> >
> >
> > We have added ablations isolating the key differences between TAIL and prior sequence-based meta-learners:
> >
> >
> > * non-causal vs causal architectures,
> > * universal feature projection vs fixed feature spaces,
> > * universal label dictionary vs fixed label embeddings.
> >
> >
> > The results show that each of these components contributes to the performance gap between TAIL and the baselines, clarifying *why* TAIL outperforms them in the reported settings.
> >
> >
> > ---

---

### Official Review · Reviewer_nB5Q · 2025-10-17

**Soundness:** 2
**Presentation:** 3
**Contribution:** 3
**Rating:** 6
**Confidence:** 5

**Summary:**

The paper proposes TAIL, a non-causal transformer trained on sequences of support examples and unlabeled query samples so that the transformer learns an implicit learning algorithm and predicts query labels in a single forward pass. TAIL extends previous approaches to generalize to unseen domains and modalities using an arbitrary number of classes.

**Strengths:**

- The extension of previous approaches such as CAML and GPICL, to multiple modalities and large number of classes is interesting and in line with the current research directions in foundation models.
- The distinction between algorithm-explicit and algorithm-implicit learning and the formal formulation of universality is useful and clearly described.
- Experimental results show the strength of the proposed approach.

**Weaknesses:**

- The formulation of the meta-learning problem and particularly the task definition could be improved by also providing references to survey papers (e.g., [1], [2], etc).
- The motivation behind the choice of the vision and text encoder should be clarified. An ablation experiment on different encoders can strengthen the paper, similarly to what has been done in [3].
- The claim about the computational efficiency at scale is not well supported. Additional information about the memory usage for training and inference and the training time for the different baselines is needed to support this claim.
- The claim of cross-domain generalization is not fully evaluated. It is unclear how data leakage between datasets (e.g., MetaAlbum or ImageNet used for training vs. Birds or Airplanes used for testing) is prevented, given overlapping semantic classes. A stronger assessment of cross-domain capabilities should consider completely different domains (e.g., training on ImageNet + MetaAlbum and testing on MedIMeta).
- Some experimental details are missing, such as how the training tasks are sampled (e.g., within or cross dataset), how the training dataset are shown to the model (e.g., randomized or in a sequence), and what are the hyperparameters used for training.

**Questions:**

- Linear Probe and ProtoHead perform similarly to TAIL (considering the standard deviation) on cross-modal generalization to unseen modalities (Tab. 4). A discussion on this is needed. Moreover, what is the benefit of using TAIL considering the much larger memory and computation requirements?
- What is the rationale behind the selection of the training dataset? Would training only on MetaAlbum be sufficient to achieve strong generalization, and what additional benefit does including ImageNet or other datasets provide?

---

> ### Author Response · Authors · 2025-11-18
>
> Thank you very much for your detailed review and for your helpful comments and questions! We are currently preparing a detailed reply addressing all of your points and will post it shortly.
>
> We noticed that in the *Weaknesses* section you cited references [1], [2], and [3], but it seems that these references were not included in your review. Would you be able to provide the bibliographic information for those works? Knowing which papers you are referring to will help us prepare a more accurate and fully informed response. Thank you!

---

> > ### Comment · Reviewer_nB5Q · 2025-11-18
> > **Missing references**
> >
> > I apologize for this. In the previous message I was referring to the following references:
> >
> > [1] Vettoruzzo, Anna, et al. "Advances and challenges in meta-learning: A technical review." IEEE transactions on pattern analysis and machine intelligence 46.7 (2024): 4763-4779.
> >
> > [2] Hospedales, Timothy, et al. "Meta-learning in neural networks: A survey." IEEE transactions on pattern analysis and machine intelligence 44.9 (2021): 5149-5169.
> >
> > [3] Fifty, Christopher, et al. "Context-aware meta-learning." arXiv preprint arXiv:2310.10971 (2023).
> >
> > Please note that these are only suggestions, feel free to not include them if you think they are not relevant.

---

> ### Author Response · Authors · 2025-11-21
> **Response to Reviewer nB5Q (R3)**
>
> Thank you for the thorough and careful review, and for your positive remarks on our theoretical formulation and empirical performance across modalities. In the following, we address your comments and questions in detail.
>
>
> ---
>
>
> ### **R3.1. References to survey papers**
> > *“The formulation of the meta-learning problem and particularly the task definition could be improved by also providing references to survey papers.”*
>
>
> Thank you for supplying the missing citations and for the suggestion. The surveys by Hospedales et al. (2021) and Vettoruzzo et al. (2024) offer useful historical and conceptual context and we therefore have incorporated them into the introduction and related work in the new revision. These surveys also support our framing of the cross-domain/meta-generalization challenge that TAIL aims to address.
>
>
> ---
>
>
> ### **R3.2. Encoder motivation and ablations**
>
>
> > *“The motivation behind the choice of the vision and text encoder should be clarified… an ablation experiment on different encoders can strengthen the paper.”*
>
>
>
>
> We agree with the reviewer that an exploration of different encoders would be an interesting avenue to pursue. Indeed, we have investigated this question in preliminary experiments and found the choice of encoder does play a key role. However, we would like to note that our core novelties are invariant to the choice of encoder. We would like to further note that all baseline methods employ the same encoder and therefore the results are comparable. In preliminary experiments we also found that changing the encoder affected all methods similarly. Ultimately we decided to use the CLIP LAION encoder due to its robust performance in our preliminary investigations. Due to the time-constraints in the rebuttal period we leave a wider exploration of potential gains from using different encoders to future work.
>
>
> ---
>
>
> ### **R3.3. Computational efficiency**
>
>
> > *“The claim about the computational efficiency at scale is not well supported. Additional information about memory usage […] and training time […] is needed.”*
>
>
> We agree.
> We are currently running an analysis of the training time and memory requirements of TAIL and the baselines. We will add a **training cost plot** (not just inference costs), an encoder vs. transformer **compute breakdown** and a discussion of memory requirements to a new revision of the manuscript next week, together with the other experimental results.
>
>
> ---
>
>
> ### **R3.4. Cross-domain leakage and evaluation**
>
>
> > *“It is unclear how data leakage between datasets […] is prevented […] A stronger assessment should consider completely different domains.”*
>
>
> This is a fair point and we appreciate this observation. To address this, we will add a detailed analysis of class overlap between the datasets, clarifications about how tasks are sampled to avoid leakage, and additional experiments involving fully disjoint domains (e.g., MedIMeta), as suggested.
> We expect to have these results next week.
>
>
> ---
>
>
> ### **R3.5. Missing details: task sampling, ordering, hyperparameters**
>
>
> > *“Some experimental details are missing, such as how the training tasks are sampled (e.g., within or cross dataset), how the training dataset are shown to the model (e.g., randomized or in a sequence), and what are the hyperparameters used for training.”*
>
>
> We thank the reviewer for pointing this out.
> We have placed the task sampling and training details already present in the experiments section in the methods section, as we think the proximity will help to make this more visible.
> We have additionally expanded the appendix with details on the task and dataset sampling strategy and the optimizer parameters.
>
>
> ---
>
>
> ### **R3.6. Linear Probe vs TAIL**
>
>
> > *“Linear Probe and ProtoHead perform similarly […] on cross-modal generalization to unseen modalities (Tab. 4). […] what is the benefit given TAIL’s larger memory and compute?”*
>
>
> Thank you for raising this point. We agree that our current figure does not effectively communicate the underlying trade-offs and currently undersells TAIL.
> For typical class counts, **TAIL is actually more memory–efficient and faster** than linear probing because it avoids per-task optimization loops.
> We have updated Figure 2 and provided a clearer discussion.
>
>
> ---
>
>
> ### **R3.7. Training dataset selection (MetaAlbum + ImageNet)**
>
>
> > *“What is the rationale behind the selection of the training dataset? Would […] MetaAlbum alone be sufficient?”*
>
>
> Thank you for the question. Indeed, we found in preliminary experiments that using a large meta-dataset is crucial for TAIL’s performance.
> Using MetaAlbum alone or ImageNet alone is insufficient. The inclusion of ImageNet provides substantially more data, while MetaAlbum and MedIMeta provide additional class diversity.
>
>
> ---

---

> > ### Author Response · Authors · 2025-12-04
> > **Follow-ups for Reviewer nB5Q (R3)**
> >
> > ### R3.3 – Computational efficiency (training time & memory)
> >
> >
> > In line with your suggestion and R1.2, we have added a more comprehensive **efficiency analysis**. This includes training time and memory usage for TAIL and all baselines. We also discuss the ratio between encoder and transformer costs. The results confirm that TAIL is more efficient than methods that require per-task fine-tuning and in most cases with a usual label cardinality is also more efficient than Linear Probing.
> >
> >
> > ---
> >
> >
> > ### R3.4 – Cross-domain leakage & evaluation on disjoint domains
> >
> >
> > We have analysed the potential **class-overlap** across MetaAlbum, ImageNet, and the evaluation datasets, clarifying which tasks potentially have overlapping semantic classes. We would like to emphasize that *all* datasets used for the cross-domain evaluation do not overlap in the tasks definition, i.e. the annotation of the data.

---

### Official Review · Reviewer_bn9A · 2025-10-31

**Soundness:** 2
**Presentation:** 2
**Contribution:** 1
**Rating:** 2
**Confidence:** 4

**Summary:**

This paper proposes TAIL, transformer-based algorithm-implicit learner, which adopts the model-based meta-learning methodology, and propose structure designs for unifying potentially distinct input domains and target class cardinalities.

**Strengths:**

The overall architecture design makes the model flexible and generalizable to various input modalities and cardinalities.

Specifically, the authors have devised random permutation mappings on top of the encoded features from the inputs, achieving the benefits of preventing the model overfitting over fixed-structured features, as well as implicitly realizing input augmentations which encourage the model robustness. Furthermore, a global learnable dictionary is employed to enable the model to adapt to test-time tasks with classes of different cardinalities.

The experiments cover both adaptation to tasks cross-domain and cross-data-modality, with the proposed method achieving the best results most of the time.

**Weaknesses:**

The proposed method is fundamentally the same as the model-based meta-learning methods that can be dated back in 2016, which the authors have identified under the Related Work section (with LSTMs or transformers): the few-shot training samples with labels are provided in together with the query sample as a sequence to the model, which directly predicts the label for the query. While the authors have identified short comings from prior works (e.g. not being invariant in sample ordering, generalizing to different modalities or class cardinalities), the novelties are strictly within the detailed architectural designs (which the reviewer has recognized as the strength of the paper). Therefore, the novelty of the paper is severely limited.

Within Section 2, the paper sets up theoretical foundations for meta-learning and so called algorithm-implicit and algorithm-explicit learning systems. However, while it is nice to formulate the notions such as practical universality, for the proposed method, there is lack of connection between these theoretical discussions in Section 2 and the designed approaches. The algorithm-implicit nature of the approach is easily recognized for the model-based approaches as the authors have surveyed. The reviewer questions the necessity of the theoretical discussions in Section 2, as well as the overall theoretical contribution of the paper.

Some parts of the write-ups need to be fixed or improved:
1. For the last sentence in the introduction section, "We demonstrate that algorithm-implicit approaches outperform algorithm, explicit ones for small support sets and varied tasks.", there is a punctuation typo: "algorithm, explicit" -> "algorithm-explicit".
2. For the first sentence under Section 2.2, the symbols shouldn't be the same for both the learning algorithm space and one algorithm within it.
3. In the last equation at the bottom of pg2 (definition of the general meta-learning problem), the notion "n" is not defined until later in Definition 3.
4. The introduction of the notion "z" at the beginning of section 3 is a bit abrupt, as the definition of the input sequence encoded based on the inputs was not made clear until later in the context.

In the experiment section, there lacks ablation studies justifying the various architectural design choices proposed in the paper.

**Questions:**

While it is perceivable that algorithm-implicit learning system has better learning capacity, as the authors have demonstrated, the reviewer wonders if the algorithm-explicit learning system has better sample efficiency in learning (where samples here mean the tests in meta-training). Have the authors conducted experiments where the meta-training tasks are limited in quantities, where the inductive bias in the algorithm-explicit learning could be the advantage?

---

> ### Author Response · Authors · 2025-11-21
> **Response to Reviewer bn9A (R2)**
>
> Thank you for the time and care you put into this review, and for noting the flexibility of our architecture and the strength of our cross-domain and cross-modality results. In the following, we address your comments and questions in detail.
>
>
> ---
>
>
> ### **R2.1. Novelty relative to model-based meta-learning**
>
>
> > *“The proposed method is fundamentally the same as the model-based meta-learning methods [dating] back to 2016. […] Therefore, the novelty of the paper is severely limited.”*
>
>
> We understand the concern.
> While the **high-level paradigm** of sequence-based meta-learning has been established many years, the community has been continually building on and improving this idea. Prior approaches in this family of techniques were limited to **toy datasets** and **single domains**, or did not demonstrate cross-domain generalization, cross-modality generalization and handling arbitrary numbers of classes.
>
>
> To our knowledge, **TAIL is the first model-based meta-learner to succeed simultaneously across domain, modality, and cardinality shifts**.
> We have made this central contribution more explicit in the revision.
>
>
> ---
>
>
> ### **R2.2. Theory–method connection**
>
>
> > *“Within Section 2, the paper sets up theoretical foundations for meta-learning and so called algorithm-implicit and algorithm-explicit learning systems. However, […] there is lack of connection between these theoretical discussions in Section 2 and the designed approaches. […] The reviewer questions the necessity of the theoretical discussions in Section 2, as well as the overall theoretical contribution of the paper.”*
>
>
> Thank you for this comment. We would like to clarify our intention with the theoretical component of the paper.
>
>
> Our goal in Section 2 is to provide a **formal foundation for practical universality and algorithm-implicit vs. algorithm-explicit learning**. To the best of our knowledge, **no prior work has offered a unified and rigorous treatment of these notions**, despite them being frequently discussed informally in meta-learning and in-context learning research. We therefore believe that this theoretical framing is genuinely valuable and helps clarify an increasingly important conceptual distinction in the field.
>
>
> That said, we fully recognize the reviewer’s point that the **connection between the theoretical framework and the concrete TAIL architecture could be made more explicit**. In the revised manuscript, we have added a short paragraph to Section 3.4 which maps each theoretical construct directly to architectural components of TAIL and shows how TAIL satisfies the algorithm-implicit learner definition.
>
>
> We appreciate this feedback and believe that the revision makes the theoretical contribution and its relevance to TAIL much clearer.
>
>
> ---
>
>
> ### **R2.3. Typos and notation issues**
> > *“Some parts of the write-ups need to be fixed or improved:… punctuation typo… symbols shouldn’t be the same… notion ‘n’ not defined… introduction of ‘z’ is abrupt.”*
>
>
> Thank you for these detailed comments. We have made corrections for the first three points, and will carefully check our manuscript for any other typos or inaccuracies
> We did not fully understand your last point regarding the introduction of the notion “z” in Section 3, and would appreciate clarification.
>
>
> ---
>
>
> ### **R2.4. Missing ablations**
>
>
> > *“[…] there lacks ablation studies justifying the various architectural design choices […]”*
>
>
> We appreciate this suggestion. While several ablations are included in the appendix, we agree that the main text does not sufficiently emphasize these analyses or isolate the contributions of individual architectural components.
> We are currently running additional ablation experiments (causal vs. non-causal attention, random projection vs random permutation, and label dictionary size). We plan to have the results for those experiments next week and will add a new revision including these results and an appropriate discussion.
>
>
> ---

---

> > ### Author Response · Authors · 2025-11-21
> > **Response to Reviewer bn9A (R2)**
> >
> > ### **R2.5. Sample-efficiency question**
> >
> >
> > > *“the reviewer wonders if the algorithm-explicit learning system has better sample efficiency in learning (where samples here mean the tests in meta-training).”*
> >
> >
> > This is a valuable point.
> > Algorithm-explicit methods (e.g., gradient descent) indeed have strong inductive biases and are more likely to perform well in situations where only limited  meta-training data is available.
> > In the extreme case when only one dataset is available, meta-learning is no longer applicable.
> > However, we believe that algorithm-implicit meta-learners with flexible input domains, such as our proposed method, will make it possible to learn from a broader domain of input data and therefore alleviate the problem of insufficient meta-training data.
> >
> >
> > A helpful analogy is the evolution from **manual feature engineering** to **deep learning**:
> >
> >
> > * In the feature-engineering era, models relied on strong human-designed biases and often performed reasonably well with limited data.
> > * Deep learning introduced far weaker inductive biases but **greater representational power**, at the cost of requiring more data to fully exploit that power.
> >
> >
> > We believe that a similar shift is now happening with *learning algorithms themselves*:
> >
> >
> > * **Algorithm-explicit** approaches (e.g., gradient-based methods) encode strong assumptions about the update rule. These biases *help* when meta-training data are scarce.
> > * **Algorithm-implicit** approaches (like TAIL) place far fewer assumptions on the learning algorithm, allowing the model to learn more flexible and powerful update rules, but **requiring more meta-training tasks**.
> >
> >
> > So we fully agree that in extremely data-limited meta-training regimes, algorithm-explicit methods may outperform algorithm-implicit ones.
> >
> > We have discussed  this trade-off and the potential of universal meta-learners more explicitly in the Discussion of the revised manuscript.
> >
> > ---

---

> > > ### Author Response · Authors · 2025-12-04
> > > **Follow-ups for Reviewer bn9A (R2)**
> > >
> > > ### R2.4 – Additional ablations
> > >
> > >
> > > As indicated in the previous response, we have expanded the ablation section. The new experiments include:
> > >
> > >
> > > * **Causal vs non-causal attention**, and
> > > * **Random projection vs extended permutation**.
> > >
> > >
> > > These results are now summarized in the main text (with full tables in the appendix) and show that each architectural choice contributes meaningfully to cross-domain and cross-modality robustness and label-space extrapolation, providing a more direct justification for the design of TAIL.
> > >
> > >
> > > ---

---

### Official Review · Reviewer_xZB2 · 2025-11-04

**Soundness:** 3
**Presentation:** 3
**Contribution:** 4
**Rating:** 6
**Confidence:** 2

**Summary:**

This paper introduces TAIL (Transformer-based Algorithm-Implicit Learner), a novel meta-learning framework designed to achieve practical universality (the ability to learn across tasks with differing feature domains, modalities, and label spaces).

The authors first develop a theoretical framework distinguishing algorithm-explicit and algorithm-implicit meta-learners, formalizing concepts such as valid learning algorithms and practical universality.

TAIL implements an algorithm-implicit learner using a non-causal transformer that directly predicts query labels from sequences of support examples and an unlabeled query. It also contains some other techs, such as universal feature encoding, etc.

Extensive experiments validate the effectiveness of TAIL from in-domain, cross-domain, cross-modality settups. It demonstrates robustness to tasks with up to 20× more classes than in training.

**Strengths:**

1. The motivation is clear and strong.

2. The paper’s claims are well-supported by both strong theoretical grounding and comprehensive empirical validation.
- The theoretical categories of algorithm-implicit vs. explicit learning are novel.
- Experiments cover a broad evaluations: in-domain, cross-domain, cross-modality, and label extrapolation, with ablation studies validating the design choices. Also, the computation is efficient.

**Weaknesses:**

1. Pretrained encoder dependency: Although random projections help, the reliance on large pretrained encoders complicates the claim of “from-scratch” universality, as part of the final performance may stem from the backbone’s prior knowledge. The importance of this encoder component should be explored more thoroughly. For example, by using the same pretrained encoder for both TAIL and the baselines to isolate its contribution.

2. Computation cost in experiments: While the paper reports efficiency results, it lacks a detailed analysis of the training cost at scale. It would also be helpful to quantify how much of the total computational cost arises from the encoder.

3. Insufficient ablation analysis: The impact of the randomized extended permutation 𝜋 could be analyzed more systematically. Table 5 in the appendix only compares results with and without 𝜋. A sensitivity analysis would clarify how strongly TAIL depends on this component.

**Questions:**

In addition to the above weakness, some additional questions:

1. Could you provide more insight into how the non-causal transformer architecture outperforms the causal transformer arch?
2. Have you compared TAIL’s performance to recent foundation-model-based few-shot learners (I just searched and found one paper attached below. To answer this question, the authors could use any other related papers or models, not necessarily the following one)
- Prompt, Generate, then Cache: Cascade of Foundation Models makes Strong Few-shot Learners, CVPR 2023
3. How would the model perform if meta-trained on mixed-modality/distribution tasks rather than purely modality/distribution data?

---

> ### Author Response · Authors · 2025-11-21
> **Response to Reviewer xZB2 (R1)**
>
> Thank you for your time and thoughtful review, and for highlighting the strength of our motivation, theoretical framing, and empirical evaluation. In the following, we address your comments and questions in detail.
>
>
> ---
>
>
> ### **R1.1. Pretrained encoder dependency**
>
>
> > *“the reliance on large pretrained encoders complicates the claim of ‘from-scratch’ universality[…]. The importance of this encoder component should be explored more thoroughly, […] by using the same pretrained encoder for both TAIL and the baselines to isolate its contribution.”*
>
>
> Thank you for raising this. We believe there may have been a misunderstanding:
> **TAIL and all baselines already use the exact same pretrained encoder**, and the universal feature encoding acts *on top* of this shared representation. We have clarified this more explicitly in the revision.
>
>
> We would like to note that TAIL itself, i.e. the transformer, is trained from scratch and does not use a pretrained model.
>
>
> ---
>
>
> ### **R1.2. Computation cost and scaling**
>
>
> > *“While the paper reports efficiency results, it lacks a detailed analysis of the training cost at scale. It would also be helpful to quantify how much of the total computational cost arises from the encoder.”*
>
>
> We agree.
> We are currently running an analysis of the training time and memory requirements of TAIL and the baselines, as well the contributions of the encoder. We will add a **training cost plot** (not just inference costs), and an encoder vs. transformer **compute breakdown** to a new revision of the manuscript next week, together with the other experimental results.
>
>
> ---
>
>
> ### **R1.3. Ablation on randomized permutation π**
>
>
> > *“The impact of the randomized extended permutation π could be analyzed more systematically. […] A sensitivity analysis would clarify how strongly TAIL depends on this component.”*
>
>
> We appreciate the suggestion, although we suspect there may have been a confusion:
> π is a randomly sampled projection of the features obtained using the encoder, and since it is **randomly resampled on each episode**, its effect cannot be analyzed by comparing different instances of π.
>
>
> To address the (perceived) spirit of the question, we will include a comparison between a standard random projection (not restricted to extended permutations), and the randomized extended permutation currently used; we will add the results to the manuscript in next week's revision.
>
>
> If we misinterpreted your intention, we would appreciate clarification.
>
>
> ---
>
>
> ### **R1.4. Non-causal vs causal transformer**
>
>
> > *“Could you provide more insight into how the non-causal transformer architecture outperforms the causal transformer arch?”*
>
>
> Empirically, non-causal attention clearly outperforms causal (shown in Tables 1, 2 and 3).
> Conceptually, enforcing a directional order on unordered support examples is inappropriate: the model should treat demonstrations as an unordered set. Moreover, the absence of a causal mask allows the representations of the support examples to **attend to each other symmetrically** regardless of their order.
>
>
> We will include an ablation experiment using a causal transformer together with this explanation in next week's revision of our manuscript.
>
>
> ---
>
>
> ### **R1.5. Comparison to foundation-model-based few-shot learners**
>
>
> > *“Have you compared TAIL’s performance to recent foundation-model-based few-shot learners [e.g.] ‘Prompt, Generate, then Cache’, CVPR 2023”*
>
>
> Thank you for pointing this out. The comparison with contemporary foundation-model-based few-shot learners is indeed relevant.
> In our opinion, that particular paper is not a good point of comparison: since it largely relies on DALL-E being able to generate images of the appropriate class, and does not seem to be tested on classes not present in the DALL-E training set, it is unlikely to work in a true few-shot scenario with previously unseen classes.
> However, we will research similar papers and add a comparison to the related work section by next week.
>
>
> ---
>
>
> ### **R1.6. Mixed-modality meta-training**
>
>
> > *“How would the model perform if meta-trained on mixed-modality/distribution tasks?”*
>
>
> We have not evaluated this setting yet, but we agree it is a natural follow-up and we believe it is likely beneficial.
> We are currently running experiments which use multiple modalities in meta-training and will add the experimental results to our revision by next week.
>
>
> ---

---

> > ### Author Response · Authors · 2025-12-04
> > **Follow-ups for Reviewer xZB2 (R1)**
> >
> > ### R1.2 – Training cost & encoder vs transformer compute
> >
> >
> > In the revised manuscript, we now include a detailed computational analysis covering both **training and inference**. This includes a plot of training time versus number of classes, memory usage for TAIL and baselines, and a discussion of the encoder and transformer components. The compute is dominated by the encoder (shared across all methods), while the additional overhead introduced by TAIL is comparatively modest. We precomputed the feature encodings and only included the actual runtime of TAIL and the baselines in our analysis.
> >
> >
> > ---
> >
> >
> > ### R1.3 – Ablation on randomized permutation π vs standard projection
> >
> >
> > We have added an ablation comparing the **randomized extended permutation** used in TAIL with a **standard random projection**. The results show that both variants perform similarly, with the extended permutation providing a small improvement. This shows that the main benefit comes from the universal randomized projection mechanism itself, and that TAIL is not overly sensitive to the specific type of projection.
> >
> >
> > ---
> >
> >
> > ### R1.4 – Causal vs non-causal transformer
> >
> >
> > The revised manuscript now includes an ablation where we replace the non-causal transformer with a **causal** one. As expected from our conceptual argument, performance degrades notably in this setting. This empirically confirms that removing the artificial ordering constraint and allowing **symmetric attention** over support examples is important for good performance.
> >
> >
> > ---
> >
> >
> > ### R1.5 – Comparison to foundation-model-based few-shot learners
> >
> >
> > We have expanded the Related Work section to discuss recent **foundation-model-based few-shot learners**. We explain why these approaches rely heavily on the underlying foundation model’s coverage of classes and modalities, and argue that they are less suitable for the strict few-shot, unseen-class setting considered in our paper.
> >
> >
> > ---
> >
> >
> > ### R1.6 – Mixed-modality meta-training
> >
> >
> > We have now run experiments where TAIL is meta-trained on **mixed-modality** task distributions, combining image and text tasks during meta-training. The revised manuscript includes these results. Empirically, mixed-modality training does not significantly improve performance. We believe this is an artifact of the composition of the meta-dataset: the comparatively small amount of text data does not significantly increase the data diversity provided by our large image-classification meta-training set. We speculate that exposing TAIL to a larger cross-modal meta-training set with a broader variety of feature domains could strengthen the learned algorithm and add robustness on cross-modality evaluation.
> >
> >
> > ---

---

### Author Response · Authors · 2025-11-21

We would like to sincerely thank all reviewers for their thoughtful and constructive feedback. Your comments have been extremely helpful in improving both the clarity and technical quality of the paper.

We have already incorporated a substantial portion of the suggestions into a new revision of the manuscript, which we are uploading together with this response. All modifications made so far are **highlighted in orange** in the revised manuscript for ease of review.

Several reviewers recommended running additional experiments (encoder analyses, computational breakdowns for training, ablations on architectural components, and cross-domain training). We fully agree with these suggestions and are currently running the corresponding experiments. We will upload another revision **next week**, including all new experimental results, and an expanded discussion analyzing their implications.

---

> ### Author Response · Authors · 2025-12-04
> **Update**
>
> Since our previous responses, we have conducted the additional experiments and analyses mentioned in the original replies (computational analyses for training, ablations on architectural components, and cross-domain training). The revised manuscript now includes these results and an expanded discussion. Across all new experiments, the conclusions are consistent with our original claims: TAIL remains competitive or superior to strong baselines while enabling cross-modality settings and label-space extrapolation.

---

### Author Response · Authors · 2025-12-04
**Author Summary to the Area Chair**

We thank the reviewers and the AC for their time and thoughtful effort in evaluating our work. We would like to provide a summary of the reviews, how we addressed all concerns raised by the reviewers, how we improved our manuscript, and why we believe our paper **"Universal Algorithm-Implicit Learning"** makes a strong contribution to the ICLR community.




---


## TOC
1. Summary of the original reviews
2. Summary of the discussion period
3. Why our method (TAIL) is a strong contribution
4. Clarity issues resolved during the rebuttal
5. Additional experiments and improvements now completed
6. Status after rebuttal
7. Broader Context and Closing Remarks


---


## 1. Summary of the original reviews


#### **Reviewer xZB2 (R1) — Score: 6**


**Positive:**


* Praised the strong motivation, clear theoretical framing, and comprehensive empirical validation.
* Highlighted the novelty of distinguishing algorithm-implicit vs. algorithm-explicit learning.


**Concerns:**


* Asked for clarification on the dependence on pretrained encoders. **[R1.1]**
* Requested more analysis of computational cost (especially training). **[R1.2]**
* Suggested more detailed ablations, particularly on the randomized permutation. **[R1.3]**
* Asked about causal vs non-causal transformer choice **[R1.4]**, comparisons to foundation-model few-shot methods **[R1.5]**, and mixed-modality meta-training **[R1.6]**.




#### **Reviewer bn9A (R2) — Score: 2**


**Positive:**


* Highlighted the flexibility of the architecture and its strong performance across domains and modalities.
* Recognized the value of the permutation mapping and label dictionary.
* Recognized SOTA results across the benchmark settings.


**Concerns:**


* Questioned novelty, suggesting TAIL was similar to earlier model-based meta-learners (2016–2018) **[R2.1]**.
* Felt the theoretical section’s connection to the method was unclear. **[R2.2]**
* Pointed out several minor writing and notation issues. **[R2.3]**
* Requested more ablations to justify architectural choices. **[R2.4]**
* Asked about sample-efficiency comparisons to algorithm-explicit methods. **[R2.5]**




#### **Reviewer nB5Q (R3) — Score: 6**


**Positive:**


* Found the theoretical framing (algorithm-explicit vs. implicit, universality) useful.
* Praised the strong empirical results, especially generalization to unseen modalities.
* Viewed the method as a meaningful extension of recent model-based meta-learners.


**Concerns:**


* Requested additional citations to meta-learning surveys. **[R3.1]**
* Asked for clearer motivation for encoder choice and encoder ablations. **[R3.2]**
* Sought more detailed analysis on computational efficiency. **[R3.3]**
* Raised concerns about cross-domain leakage and asked for stronger evaluations on entirely disjoint domains. **[R3.4]**
* Requested more clarity on task sampling and hyperparameters. **[R3.5]**
* Asked for clarification of the trade-off between Linear Probe/ProtoHead and TAIL. **[R3.6]**
* Asked about training dataset selection **[R3.7]**


#### **Reviewer RReC (R4) — Score: 2**


**Positive:**


* Appreciated the clarity of the theoretical exposition.
* Highlighted the breadth of experiments and the SOTA results.


**Concerns:**


* Claimed the architecture and theoretical contributions were “almost identical” to two specific prior works. **[R4.1]**
* Noted missing related work on ICL mechanisms. **[R4.2]**
* Found the algorithm-explicit vs implicit distinction insufficiently defined. **[R4.3]**
* Requested more ablation studies to explain why TAIL outperforms prior transformer-based meta-learners. **[R4.4]**
* Pointed out typos. **[R4.5]**


---


## 2. Summary of the discussion period


* The reviews were released on 12th of November
* We provided responses to all reviewers and an initial revision of our manuscript on 21st of November
* Unfortunately the reviewers did not engage before the PCs closed the discussion on 28th of November

---

## 3. Why our method (TAIL) is a strong contribution

**Our proposed method (TAIL) is the first model-based meta-learner that succeeds simultaneously in:**


* **cross-domain generalization (transfer to unseen datasets with unseen semantic classes)**,
* **cross-modality generalization (transfer to an unseen modality, e.g. image -> text)**, and
* **label-space extrapolation (20× more classes than seen during training).**


This combination has not been demonstrated by any prior model-based meta-learner (including SNAIL, CAML, GPICL). Our universal feature encoding strategy using randomized extended permutations enables tasks with *arbitrary feature domains* and our random injection label embedding enables tasks with *arbitrary label spaces*. Both are novel mechanisms improving upon previous works, which cannot handle these settings due to fixed feature dimensions or fixed label embeddings.


All **reviewers agreed that the empirical results are strong, broad, and relevant** to the direction the field is moving.

---

> ### Author Response · Authors · 2025-12-04
> **Author Summary to the Area Chair**
>
> ## 4. Clarity issues resolved during the rebuttal
>
>
> Several of the initially mentioned weaknesses were based on misunderstandings that we clarified:
>
>
> **(a) “TAIL uses a different pretrained encoder than baselines.” (R1.5)**
>
>
> We clarified in both the response and the manuscript that **TAIL and all baselines use *the exact same* pretrained encoder**, so the encoder choice does not inflate TAIL’s gains. The novelty lies entirely in the universal projection + label dictionary.
>
>
> **(b) “There is no connection between the theory and the method.” (R2.2)**
>
>
> Contrary to the reviewer’s comments, both our method design and the evaluation rely directly on the theory laid out in Section 2. However, we acknowledged that the connection could have been made more explicit. We added a short subsection (Section 3.4) explicitly analyzing how the architecture of TAIL fulfills the theoretical constructs (valid algorithms, algorithm-implicit learning, practical universality), addressing this concern.
>
>
> **(c) “TAIL is almost identical to earlier works (limited novelty).” (R4.1)**
>
>
> We provided a detailed comparison showing why prior sequence-based methods *cannot* handle heterogeneous features or arbitrary label spaces and clarified how TAIL addresses these issues. Our work also introduces the first formalization of *practical universality*, which multiple reviewers considered a valuable contribution.
>
>
> ---
>
>
> ## 5. Additional experiments and improvements now completed
>
>
> All reviewers asked for additional analyses. We have now run the necessary experiments to address **all of those requests** and have added the following results in the new revised manuscript:
>
>
> * **training-time and memory-cost analysis**, including encoder vs transformer breakdown discussion (Section 5.2),
> * **causal vs non-causal transformer ablation** (Appendix C),
> * **random-projection vs extended-permutation ablation** (Appendix C),
> * **mixed-modality meta-training experiments** (Appendix C),
> * **cross-domain leakage analysis** (class overlap across datasets, Section 5.2),
> * **expanded task-sampling and hyperparameter details** (Appendix E).
>
>
> Across all additional analyses, the conclusions **remain consistent** with our initial claims: TAIL is competitive or superior to strong baselines, while enabling the more ambitious and practically relevant cross-domain/cross-modality/label-extrapolation setting.
>
>
> Moreover, some reviewers requested that we compare our work to additional lines of ** related work**.
> We incorporated all suggested related work, including the recent ICL-mechanism papers, into a concise discussion in the revised related-work section.
>
>
> **We would like to thank** all reviewers for their comments, questions and suggestions which ultimately helped improve our paper substantially. Their high-quality reviews were an integral part of the process of improving the manuscript.
>
>
> ---
>
>
> ## 6. Status after rebuttal
>
>
> * **R1 & R3**: Initially both gave a score of 6 and highlighted strengths in theory and experiments. **All** of their concerns were **addressed fully**.
> * **R2 & R4**: Had novelty concerns (e.g. “almost identical” to GPICL/CAML). We clarified the architectural and theoretical distinctions from previous work and moreover **fully addressed all** their other questions.
>
>
> Importantly, **none** of the reviewers questioned the empirical strength or relevance of the task setting. Concerns were mainly about presentation and clarity or lack of novelty, which have now been fixed. All suggestions for additional ablation studies or additional references have been implemented.
>
>
> ---
>
>
> ## 7. Broader Context and Closing Remarks
>
>
> The question of **how to build meta-learners that generalize across domains, modalities, and label spaces** is central to the future of few-shot learning and intersects with current trends in foundation models and in-context learning. Our paper presents:
> * a simple and scalable architecture,
> * a unified theoretical framing,
> * strong empirical validation,
> * the first demonstration of practical universality in cross-domain, cross-modality meta-learning.
>
>
> The revised manuscript is **significantly clearer, and experimentally stronger** than the original. We have incorporated all recommended references, clarified theoretical definitions and their relationship to the architecture, added missing implementation details, and completed the additional computational and ablation studies that reviewers requested.
>
>
> We hope that the AC will take into account that the initial concerns have been **fully resolved** in their final evaluation of our contribution and thank the AC again for their time and consideration.
>
>
> ---

---

### Meta-Review · Area_Chair_bbtm · 2025-12-20

**Summary:**

This paper presents a method, referred to as TAIL, which shares the same spirit with the model-based meta-learning but steps forward to succeed simultaneously in: (1) cross-domain generalization; (2) cross-modality generalization; (3) label-space extrapolation.  Reviewers agree that the method is sound and empirical results are good. Some of critical concerns raised by reviewers are summarized below.
1. Experiments for more ablation study and more empirical comparison.
2. Comparison in terms of computational efficiency.
3. Theoretical section’s connection to the method is unclear.
The authors did a nice job in responding to reviewers' concerns, highlighting what's been changed in the revision. To my best knowledge, this paper seems to be the first work on a model-based meta-learning which generalizes across modalities/domain and performs label-space extrapolation. The paper claims that the TAIL represents a structurally universal function but it does not prove it is a universal learner.
In other words, it does not prove that TAIL achieves practical universality (convergence/consistency). The paper proves specific structural properties (invariances and equivariances) that are necessary conditions for universality.  The authors need to make the validity to be clear,
via either referring to existing results (if any) or providing its proof.

**Reviewer Concerns:**

Most concerns were addressed satisfactorily. Regarding R2.2, however, it remains unclear whether validity is assumed or theoretically proven.

**Reviewer Scores:**

Since the authors did a nice job, score could be raised a little bit, but not up to the threshold.

---

### Decision · Program_Chairs · 2026-01-26

Reject